# Development of an open-hardware semen homogenizer and application to serotonin effects on sperm motility

Nancy A. Juárez-Contreras[1], Cindy U. Rivas-Arzaluz[1¤], Oscar Gutiérrez Pérez[2],
María E. Ayala[3], Andrés Aragón-Martínez[1]*

**1** Gamet and Technological Development Laboratory, Biomedicine Unit, Faculty of Higher Studies Iztacala, National Autonomous University of Mexico (UNAM), State of Mexico, Mexico City, Mexico, **2** Center for Teaching, Research and Extension in Swine Production (CEIEPP), Faculty of Veterinary Medicine and Zootechnics, National Autonomous University of Mexico (UNAM), State of Mexico, Mexico City, Mexico, **3** Biology of Reproduction Unit, Puberty Laboratory, Faculty of Higher Studies Zaragoza, UNAM, Mexico City, Mexico

¤ Current address: Department of Developmental Genetics and Molecular Physiology, Institute of Biotechnology, National Autonomous University of Mexico (UNAM), Cuernavaca, Morelos, Mexico, ZIP 62210.
* andres.aragon@iztacala.unam.mx

## Abstract

Proper homogenization of semen samples is crucial for assisted reproductive technology (ART) and analysis of sperm motility kinematics in experimental research. Accurate homogenization ensures a consistent number of sperm per insemination in ART and uniforms number of sperm per treatment in research. Homogenization is particularly important when sperm are treated with exogenous compounds that must be evenly distributed throughout the experiment. Traditionally, semen samples are homogenized manually with a "gentle" approach or with commercial devices that are not specifically designed for this purpose. In this study, we designed, constructed, and validated an open-hardware semen homogenizer for sperm motility analysis. For validity, the hardware was used to investigate the effects of serotonin (5-HT) on boar sperm kinematics at the subpopulation level using a computer-assisted sperm analysis (CASA). All components of the homogenizer are readily available, including the STL files for the 3D-printed parts, firmware, assembly instructions, and operation guidelines. Objective homogenization was evaluated by measuring sperm density at different numbers of homogenization cycle, while "gentle" homogenization was assessed based on manual duration and inclination angle. Sperm motility kinematics showed no changes at the average level, however distinct effects emerged within subpopulations. Exposure to 5-HT differentially affected sperm motility kinematics across subpopulations displaying no change, increase or decrease in velocity indices. Under over-homogenization conditions, a clear interaction between prolonged homogenization and 5-HT exposure was observed in all subpopulations. Results suggest that objective homogenization is essential to ensure consistent sperm density,

**Data availability statement:** Data are full available in https://doi.org/10.17605/OSF.IO/ESFWG.

**Funding:** AAM received financial support from DGAPA-PAPIIT number IN221018, IT201021 and IN224925. The funders had no role in study design, data collection and analysis, decision to publish, or preparation of the manuscript.

**Competing interests:** The authors have declared that no competing interests exist.

whereas over-homogenization alters sperm responses, potentially leading to biological misinterpretations.

## Introduction

In our lab semen quality is routinely evaluated with aid of open-hardware. In this context, glass slides and plastics are heated on a thermal plate constructed with the hot bed of a 3D printer, sperm motility kinematics are analysed using an open-source computer-assisted sperm analysis system (CASA) [1], and sperm density and vertical velocity are evaluated in the UPSPERM device [2]. Homogenization of semen ensures uniform sperm density, which is essential for both the reliable operation of fertilization assays or CASA systems. Obtaining a representative sperm sample is fundamental for statistical analysis, as it ensures that all sperm have an equal probability of being selected for measurement.

In most laboratories, homogenization of semen is typically performed in a tube by means of manual operation using repeated forward and backward rotational movements (See the S1 Video and the video Hand_homogenization.mp4 in videos section of https://doi.org/10.17605/OSF.IO/ESFWG). This process is often described as "gentle homogenization" in scientific reports [3,4]. Gentle homogenization is necessary because vigorous agitation may cause damage to the sperm cells or collisions between sperm and the tube walls [5–8]. Human factors affect semen homogenization as different technicians may perform varying numbers of rotational movements at different speeds that may potentially cause damage to the sperm cells.

A key application of semen homogenization lies in the storage and handling of semen samples. Among the commercial devices used to "blend/mix/agitate/rock" samples are orbital mixers/shakers, rotational mixers, nutating mixers, reciprocating mixers, and hand-motion shakers (https://www.coleparmer.com/c/shakers). These proprietary devices are multi-purpose and not specifically designed for semen homogenization, but they can be adapted to simulate hand movements. For example, the Multi Bio RS-24 (https://www.boeco.com/mixer-stirrer-and-rotator/boeco-rotator-multi-bio-rs-24&sk=78, approximately 675 dollars before VAT in México) is described as a programmable rotary agitator, which allows adjustment of rotation speed, angle, reciprocal rotation, and time interval. However, the number of inversions and recoils must still be determined by eye and experimentation. Another option potentially adaptable to gentle homogenization is the Cole-Parmer WA-200 Hand Motion Flask Shaker (https://www.coleparmer.com/i/cole-parmer-wa-200-hand-motion-flask-shaker-230-v/5190002) (2550.00 dollars at October 2024). This device, however, has a restricted rotation angle and high velocity (80–800 RPM). According to the manufacturer, the hand Motion Flask Shaker is intended for prolonged and continuous shaking, making it less suitable for handling sensitive biological samples such as semen.

To avoid sampling bias, the World Health Organization (WHO) recommends mixing semen—mechanically or manually—before taking aliquots [9]. A portable open-hardware laboratory rotator, mixer and shaker designed for low-capacity tubes, arranged in a mirrored configuration, has been reported previously [10]. One version

of that device provides continuous 360° rotation without velocity control, whereas the other incorporates a programmable microcontroller to regulate the rotation angle within a limited inclination range. These devices are intended primarily for mixing chemical solutions, not for handling sensitive biological samples such as semen. Moreover, both versions rely on DC motors, which require additional electronic components to regulate velocity and angular movement with precision.

Despite the lack of standardization in semen homogenization protocols, the manual movement of the tube remains a consistent and fundamental practice. We define full homogenization of the sperm suspension as the point at which no further changes in sperm concentration are observed after the homogenization procedure has been performed. Homogenization of semen samples typically involves rotational forward and backwards motion, however duration of motion may affect the results, the inclination angle remains undefined, and no criteria exist for determining when the process should stop. In practice, homogenization is usually concluded based on visual inspection or after an arbitrary period.

The design and construction of an open-hardware device specifically intended for semen homogenization represents a feasible alternative to modifying commercial equipment or adapting open-hardware mixers and shakers that were not originally designed for semen homogenization. Proper re-suspension of semen is particularly critical when evaluating the effects of exogenous agents on sperm function, because uneven mixing can confound exposure and bias motility readouts. For the case of serotonin (5-HT), sperm has been reported to express proteins related to serotonergic communication, including receptors, transporters, and metabolizing enzymes [11]. As an example, an increase in sperm velocity has been reported when human non-capacitated sperm was treated with 5-HT. Other studies have reported changes in average motility parameters of hyperactivated hamster sperm exposed to 5-HT receptor agonists or antagonists [12,13]. Despite these findings, the structure of kinematic subpopulations of sperm exposed to 5-HT remains unknown. A novel computational approach previously identified alterations in the kinematic subpopulation structure of boar sperm exposed to ketanserin (antagonist of the $5\text{-HT}_2$ receptor family), with results showing increased velocity accompanied by reduced linearity [14]. None of these studies addressed the problem of semen sample homogenization.

The procedure for exposing sperm to substances such as 5-HT requires a previously homogenized semen sample. Once the substance is added, the semen is homogenized but the substance is not; therefore, the sample must be homogenized again to ensure uniform exposure. We refer to this additional homogenization step as over-homogenization, defined as the condition in which a semen sample is mixed beyond the level required for standard homogenization. Whether this additional homogenization step affects kinematic outcomes remains unknown.

The objectives of this study were (1) to design and construct an open-hardware device capable of inverting and recoiling at defined inclination angles and velocities, and repeating the procedure a predetermined number of times. This device was named the "Customizable Automated Semen Homogenizer" (CASHo). (2) As a proof-of-concept we use CASHo to investigate the effect of potential over-homogenization on the structure of kinematic subpopulations in sperm exposed to 5-HT in a non-capacitating medium.

## Materials and methods

### Ethics statement

Semen doses intended for artificial insemination were donated by the CEIEEP. Thus, this work not involved animals.

### Hardware description and operation instructions

The CASHo was designed to provide homogeneous mixing and reduce operator variability during the handling of cell suspensions. The system features a fully customizable structure, encompassing both the aluminum framework and the firmware, allowing users to adjust the device according to specific laboratory requirements. Its tube holder was conceived to accommodate tubes of different capacities, ensuring versatility across experimental setups. In addition, the rotation speed, angle, and number of cycles can be configured through the firmware, offering precise control over the mixing process. The

design is also scalable, permitting expansion of the tube holder to process a higher number of samples when needed. Although developed primarily for semen homogenization, the CASHo was intended to enable objective mixing of other types of cellular suspensions where reproducible mechanical agitation is desirable.

CASHo consists of an extruded aluminum base and a set of 3D-printed components that support a stepper motor and custom-made parts designed to hold a 50 mL tube. The electronic control board is an Arduino UNO, which receives instructions via Arduino App (https://www.arduino.cc) through a computer serial monitor. The tube holder performs an inverted pendular motion at a 180-degree angle, a movement selected because it replicates the hand motion during manual homogenization (see the S1 Video and the video Hand_homogenization.mp4 in the Videos section at https://doi.org/10.17605/OSF.IO/ESFWG).

Commercial multipurpose devices available on the market can regulate movement speed but do not reproduce the motion typically performed by technicians during homogenization. By contrast, CASHo accounts for both speed and motion of hand-operated techniques. The device enables the application of objective definitions of homogenization and "gentle homogenization."

CASHo cost represents approximately 22% of the least expensive commercial option and 6% of the most expensive device currently available. All components are low-cost and readily accessible through e-commerce platforms. A detailed list of components, with reference links, prices and suppliers, is provided in S1 File and in Supplementary Information at https://doi.org/10.17605/OSF.IO/ESFWG. Unlike commercial devices, which generally allow adjustments of speed and angle but not a defined number of cyclic movements, CASHo enables precise modification of the number of cycles, rotation angle, and speed through firmware adjustment. This flexibility allows adaptation to semen homogenization across different species. The devise may also be adapted for homogenizing other cell suspensions, although technical parameter adjustments and validation of effectiveness are required on a case-by-case basis.

Several components of CASHo were designed for 3D printing using OpenSCAD (https://openscad.org) (Table 1). The design files (STL), code source (scad), and supporting information are in the STL section at https://doi.org/10.17605/OSF.IO/ESFWG; whereas details on design files, components and mounting details are in S1 File and in https://doi.org/10.17605/OSF.IO/ESFWG.

The S1 Video demonstrates the proper hand positioning for holding a tube containing a semen sample, along with the angular forward and backward movements used during manual homogenization; whereas the S2 Video (also available as CASHo_homogenization.mp4 in the Videos section at https://doi.org/10.17605/OSF.IO/ESFWG) presents the homogenization procedure using the fully assembled CASHo device connected to a computer via USB. The video shows the Arduino serial monitor with instructions for initiating a work routine, as well as real-time output displaying the number of cycles and the duration of each cycle. The firmware file Homogenizador_v6.ino (available at https://doi.org/10.17605/OSF.IO/ESFWG) is used to control the device; it includes variable definitions and specifies the pin configuration required to interface with the stepper motor driver.

**Table 1. Design files of the 3D-printed parts of CASHo.**

| Design file name | File type | Open source license |
|---|---|---|
| Junction_piece | STL, scad | GNU GPLv3 |
| Rod_adapter | STL, scad | GNU GPLv3 |
| Tube_holder | STL, scad | GNU GPLv3 |
| Nema_support | STL, scad | GNU GPLv3 |
| Rod_support | STL, scad | GNU GPLv3 |
| Hand_operated_homogenization | mp4 | GNU GPLv3 |
| Device_homogenizing_process | mp4 | GNU GPLv3 |
| Agitador_v6 | ino | GNU GPLv3 |

Detailed operating instructions for CASHo are available on protocosl.io (https://dx.doi.org/10.17504/protocols.io.4r3l29ew3v1y/v1).

### Pre-Safety checks

Ensure that the aluminum base and all parts are securely fastened. Confirm that all wiring is properly connected and tightly secured. Verify that the power supply is switched off.

Launch the Arduino IDE on the computer, open the file Homogenizer_v6.ino, and verify communication between the IDE and the Arduino UNO board. Uploading the firmware is required only once, unless modifications are made to the code in Homogenizer_v6.ino.

Ensure that the Tube_holder is in vertical position, which represents the original starting point.

Switch on the power supply to activate the stepper motor driver. Hold the Tube_holder to confirm that it remains stable, and insert a tube containing diluted semen.

Open the serial window in the Arduino IDE and verify that communication is established at 9600 baud. Enter "1" or "2" in the input section of the serial window to begin the routine. The motor shaft will then rotate, producing movement of the Tube_holder.

The output section of the serial monitor window displays the number of cycles defined in the routine, as well as the duration of each cycle in seconds (S2 Video and CASHo_homogenization.mp4 at https://doi.org/10.17605/OSF.IO/ESFWG).

### Homogenization and over-homogenization of semen

The initial step consisted of determining, with CASHo, the number of cycles required to achieve homogenization in fresh semen (within five hours of collection) and refrigerated semen (after 24 hours of storage). A semen sample was considered homogenized when sperm concentration in an aliquot did not significantly change after subjecting the sample to varying numbers of cycles in CASHo. A semen sample was considered over-homogenized when the number of mixing cycles applied in CASHo exceeded the level required to achieve homogenization.

### Semen collection, handling, and storage procedures

Semen samples (N = 5 boars, ranging in age from 1.8 to 2.3 years) were collected at Centro de Enseñanza, Investigación y Extensión en Producción Porcina (CEIEEP), Facultad de Medicina Veterinaria y Zootecnia (UNAM), using the gloved-hand technique and diluted with Androstar Plus extender (Minitube, Tiefenbach, Germany) and subsequently transferred to the laboratory.

Semen samples were (1) subjected to experimental procedures on the same day (within five hours of collection) or (2) used after 24 hours of storage. In the latter case, samples were stored at 17 °C in a temperature-controlled chamber, with the tubes placed horizontally. Commercial semen extenders allow the preservation of boar semen for several days [4]; therefore, samples were stored at 17°C [1,5,6], as this temperature has been reported to provide slight benefits for the metabolic activity and membrane integrity of boar sperm [15]. During storage, sperm cells tend to sediment, making re-suspension necessary prior to use. Examples of diluted boar semen stored in tubes placed vertically or horizontally for 24 hours are provided in the Figures section at https://doi.org/10.17605/OSF.IO/ESFWG.

The donor boars at CEIEEP were selected based on proven fertility and are routinely used in genetic dissemination programs for regional pig producers, as well as for pig production at the center's experimental farm. A strict management protocol is followed, including weekly semen collections to ensure consistent semen quality in terms of volume and concentration. Collections with progressive motility (evaluated subjectively) below 80% or with more than 15% morphological abnormalities are routinely discarded.

## Assessment of sperm concentration

Five samples, one of each of five boars were split in six aliquots. Each aliquot was subjected to 0, 1, 2, 3, 4, or 5 routines of five cycles each, corresponding to 0, 5, 10, 15, 20, or 25 total cycles. After each set of cycles performed with the CASHo device, sperm concentration was evaluated. Thus, each boar served as a block, and the experimental unit was the aliquot within boar. A 100 μL aliquot was collected with a micropipette by immersing the pipette tip halfway into the 50 mL tube. Each aliquot was placed at the bottom of a spectrophotometer cuvette (2.5 mL of capacity) containing 800 μL of Androstar Plus extender. At this stage, a heterogeneous sperm suspension was generated; homogenization within the cuvette was achieved by pipetting the full volume three times, with the beveled tip positioned at the bottom of the cuvette. Absorbance was then measured at seven distinct heights within the cuvette using the UPSPERM device, and sperm concentration was calculated [2]. Semen samples were then stored at 17°C for 24 hours, after which the same procedure of homogenization and sperm concentration evaluation was repeated.

## The 5-HT effects on sperm motility kinematics after distinct number of cycles

For the 5-HT exposure experiment, each semen sample (N = 5) was split into two aliquots of 1000 μL each. One aliquot per sample was assigned to the control group (Vh) and the other was treated with 5-HT at a final concentration of 10 μM [16]. Each boar provided one biological replicate per treatment, and aliquots derived from the same boar were considered technical replicates. After the initial homogenization of the samples, 5-HT was added to the treated aliquots, which required additional mixing to ensure proper distribution of the additive. Control samples were mixed for 5 cycles, whereas the remaining aliquots were over-homogenized for 10, 15, 20, or 25 homogenization cycles. Sperm motility was evaluated immediately after completing the assigned number of cycles.

## Assessment of sperm motility

Sperm motility was evaluated as previously stated [1]. Briefly, 15 μl of sperm suspension was placed on a slide, covered with a clean 22 × 22 mm coverslip, and examined using a B3 Clinilab phase-contrast microscope (Motic, British Columbia, Canada) equipped with a customized stage prewarmed to 38 °C. All glassware and plastic materials were prewarmed to 38 ºC on an open-hardware heat plate.

For each sample, three to four image sequences from separate fields at 100 × magnification (approximately 200 sperm per sample) were acquired using a Stingray F-033B camera (Allied Vision Technologies Inc., Exton, PA, USA) and stored digitally. Image sequences were captured with μManager software version 1.4 [17] at 60 frames per second (60 Hz) for two seconds. Image sequences were analysed using ImageJ software version 1.50d [18] and the CASA plugin [1]. Sperm average path velocity (VAP, μm/sec), curvilinear velocity (VCL, μm/sec), straight-line velocity (VSL, μm/sec), beat cross frequency (BCF, Hz), linearity (LIN, VSL/VCL), straightness coefficient (STR, VSL/VAP), amplitude of lateral head displacement (ALH, μm), and wobble (WOB, VAP/VCL) were analysed. Kinematic data for individual sperm were used to construct the dataset, which is available on the OSF platform (https://doi.org/10.17605/OSF.IO/ESFWG).

## Statistical analysis

Assumptions such as normality of the data for all variables measured in this study were checked by the Shapiro–Wilk test and residuals. Mean of sperm concentration at each time point was compared with values at 0 or 5 cycles using one-way ANOVA followed by Dunnett's test. Statistical significance was set at $P < 0.05$. At five cycles, 5-HT effects on sperm motility kinematics was assessed separately using a Student's t-test. Subsequently, the five-cycle condition was used as the reference level in the mixed-effects model, fitted to account for potential variability between boars, to evaluate over-homogenization effects across 10, 15, 20, and 25 cycles. The model was defined as

$$Y_{ijk} = \beta_0 + \beta_1 \cdot (Treatment)_j + \beta_2 \cdot (Cycles)_k + \beta_3 \cdot (Treatment \cdot Cycles)_{jk} + \upsilon_i + \epsilon_{ijk}$$

where Y$ijk$: Is the observed value of a kinematic variable (e.g., VCL, VSL) for the $i$-th boar, under treatment $j$ and homogenization cycle $k$. $\beta_0$: Intercept. Estimated mean response for the vehicle group at 5 cycles (reference level). $\beta_1$: Main effect of treatment. Difference between serotonin and vehicle across all cycles. $\beta_2$: Main effect of homogenization cycles; effect of increasing the number of homogenization cycles (e.g., 5, 10, 15, etc.). $\beta_3$: Interaction effect; how the effect of serotonin changes depending on the number of homogenization cycles. $v_i$: Random effect for boar; capturing biological variability between boars (Boar_ID). $\epsilon_{ijk} \sim N(0,\sigma^2)$: Residual error, representing variability within boars and within treatment-cycle combinations.

To identify sensitivity of kinematic subpopulations to 5-HT and to the number of homogenization cycles, the method described in [14] and the protocol for identification of kinematic subpopulations (https://dx.doi.org/10.17504/protocols.io.6qpvr9e6zvmk/v1) were followed. Briefly, individual values of motility kinematics were used as input for principal component analysis (PCA). Principal components (PC) with eigenvalues greater than one were then used as input for hierarchical clustering analysis. The number of clusters (subpopulations) was determined based on inertia gain (total variance gain).

The same mixed-effects model used to account for potential variability between boars was applied to motility parameters within each cluster. Significant main effects of treatment (Vh or 5-HT), number of homogenization cycles (10, 15, 20, or 25), and their interaction were obtained. A threshold of $P < 0.05$ was considered significant. When a significant interaction was detected, the models were further analyzed by performing separate one-way ANOVAs for cycle level. Conversely, if the interaction was not significant, the main effects of cycles and 5-HT were evaluated directly. Results are presented as mean ± standard error of motility parameters within each subpopulation, and P-values for the effects of treatment, number of cycles, and their interaction are presented as heatmaps.

Pearson's chi-square test was applied to assess whether the distribution of sperm across kinematic clusters differed significantly between treatments. For this analysis, a two-way contingency table was constructed with Treatment and Cluster as categorical variables. To evaluate the joint distribution of sperm across clusters, treatment conditions, and cycle stages, a hierarchical mosaic plot was generated. This visualization was based on a log-linear model assuming complete mutual independence among the three categorical variables. Shading of the mosaic tiles reflected standardized Pearson residuals, highlighting deviations between observed and expected frequencies under the independence model [19]. A spacing argument was incorporated to improve interpretability by visually separating the levels of each factor. All statistical analyses were performed using R software version 3.4.4 [20] on a MacBook running macOS version 15.0.1.

## Results

### Validation of CASHo

The inclination angle and cycle duration were empirically determined by recording and analyzing videos of manual homogenization performed under gentle mixing conditions. These recordings (S2 Video and Hand_homogenization.mp4 at https://doi.org/10.17605/OSF.IO/ESFWG) were used to estimate the average angular displacement and the time required to complete one forward–backward movement of the tube. These empirical observations provided the rationale for defining the standard CASHo cycle.

### Verification of angle of rotation

A clock face was printed on paper and affixed to the Nema_support using common glue. A 3D-printed ring with an arrow was attached to the flexible aluminum coupling, with the arrow positioned at 12:00 on the clock face. During operation, the angular movement of the arrow was visually verified to align with the clock face at 12:00 (original position), 03:00 (initial position), and 09:00 (S2 Video).

## Verification of velocity

Functionality of CASHo was verified by analyzing the duration of cycles and routines. The firmware logic is available in the S1 File. A 50 mL tube containing 45 mL of diluted semen was placed in the Tube_holder; these samples were used exclusively for verification and not for subsequent experiments. Cycle durations were recorded directly from the serial monitor output, using 25 consecutive cycles to exceed the number typically required for manual homogenization. The mean cycle duration was 4909 ± 0.2949 milliseconds, with a coefficient of variation of 0.0060. The dataset is provided in the S2 File and in Data section at https://doi.org/10.17605/OSF.IO/ESFWG.

## Effect of cycles of CASHo on sperm concentration

Sperm concentration stabilized after five cycles and remained at a plateau through 25 cycles in both semen stored for less than 5 hours and semen stored for 24 hours (Fig 1). In samples analyzed within 5 hours of collection, sperm concentration remained unchanged relative to the 0-cycle control (P = 0.162, 0.155, 0.149, 0.123, and 0.140 for 5, 10, 15, 20, and 25 cycles, respectively). Given the low concentration values and absence of significant variation, five cycles were considered sufficient for homogenization of semen stored for less than 5 hours.

For samples stored for 24 hours, sperm concentration reached a plateau after 5–10 cycles. Significant differences were detected compared with 0 cycles (P = 0.019, 0.002, 0.003, 0.001 and 0.002 for 5, 10, 15, 20 and 25 cycles, respectively). Comparisons with 5 cycles showed no significant differences (P = 0.959, 0.978, 0.906, 0.933 for 10, 15, 20 and 25 cycles, respectively). Thus, five cycles were also considered sufficient for homogenization of semen stored for 24 hours. These findings indicate that CASHo achieves effective homogenization with minimal cycles, making the procedure suitable for application in semen dose production.

## Effects of 5-HT and homogenization cycles on sperm motility kinematics

Once CASHo was validated and the conditions for objective homogenization were established (five cycles for 4.9 +/- 2.9 s with an inclination angle of 180°), this protocol was applied to investigate the effects of 5-HT on sperm motility kinematics.

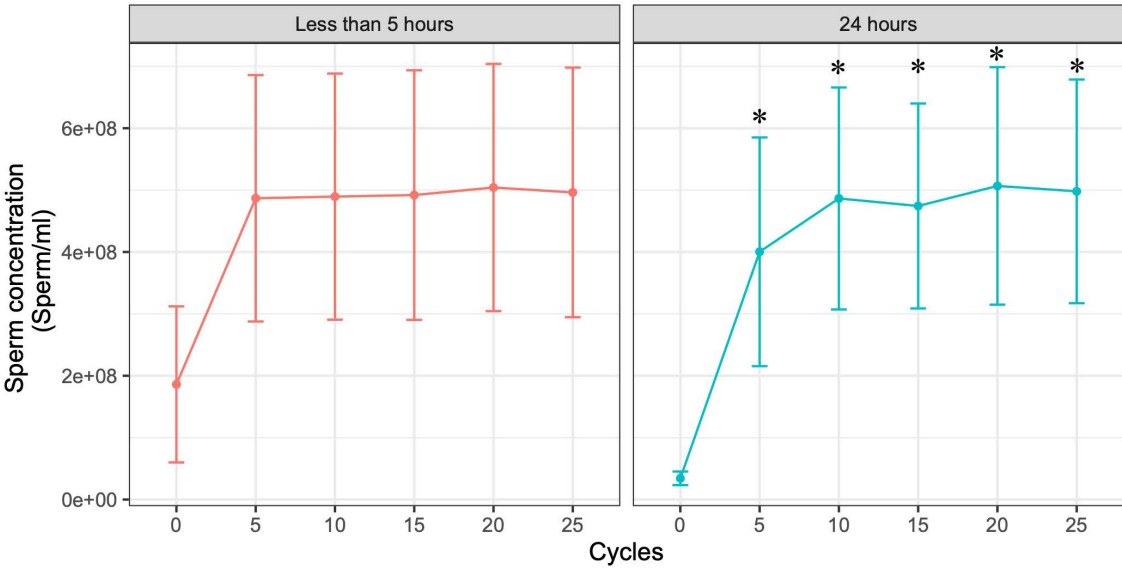

**Fig 1. Sperm concentration in semen samples stored for less than five hours or after 24 hours of refrigeration.** Sperm density was evaluated with UPSPERM after different numbers of mixing cycles in CASHo. Each value in the plot represents the average of five semen samples. Data are mean +/- SD (N = 5). Asterisks indicate differences versus 0 cycles (p < 0.05, one-way ANOVA followed by Dunnett's test).

After the semen samples were homogenized for five cycles, 5-HT was added, and samples were subsequently homogenized for 5, 10, 15, 20, or 25 cycles to mix the additive. The treatment with five cycles served as the reference condition; whereas the additional cycles (10–25) represented over-homogenization.

Two statistical analysis were employed to evaluate (a) the effects of 5-HT and homogenization cycles on the average values of motility parameters, and (b) the effects within specific kinematic subpopulations.

Exposure to 5-HT significantly affected the percentage of total motility ($p = 0.037$) and progressive motility ($p = 0.016$) (Fig 2). At five cycles average values of most kinematic parameters were unaffected ($p = 0.310, 0.421, 0.686, 0.650, 0.213$, and $0.659$ for VCL, VAP, VSL, LIN, STR, WOB, and ALH, respectively). However, a significant effect was found for BeatCross ($p = 1.89e\text{-}05$) (Fig 3).

Under conditions of over-homogenization significant effects of 5-HT, cycles, and their interaction were observed for all motility parameters but STR (Fig 3). VCL was only affected at 15 cycles.

## Effects of 5-HT and homogenization cycles on sperm kinematic subpopulations

A dataset comprising 18,644 sperm was generated, and the motility parameters of each spermatozoon were used as input for subpopulation identification. Principal component analysis (PCA) revealed that PC1 and PC2 together explained 78.71% of the variance (Table 2). Individual kinematic parameters displayed distinct associations with specific principal components (Table 2; Fig. 4A). The scores for PC1 and PC2 were subsequently used as input for hierarchical clustering. Based on the inertia gain criterion (within-cluster variance), three clusters were identified (Fig. 4B).

The number of sperm in Clusters 1, 2 and 3 was 7,840, 6,611, and 4,193, respectively. The proportions of sperm in the distinct clusters, treatment, and cycles are presented in Fig 5. A Pearson's chi-square test was conducted to evaluate the independence between 5-HT treatment and sperm subpopulation membership. The test indicated a significant deviation from independence ($p = 0.031$) (Fig 5A). Examination of standardized residuals identified Cluster 3 as the primary contributor to the observed effect. Under 5-HT treatment, the proportion of sperm assigned to Cluster 3 was significantly higher than expected ($z = 2.62$), while the vehicle group exhibited a significantly lower-than-expected proportion in the same cluster ($z = -2.62$). The standardized residuals for Clusters 1 and 2 were below the threshold for statistical relevance ($|z| < 2$). These results support the presence of a treatment-dependent redistribution of sperm among kinematic clusters, with 5-HT stimulation increasing the representation of Cluster 3.

A hierarchical mosaic plot was constructed to examine the joint distribution of sperm across kinematic clusters, treatment, and cycles (Fig 5B). The plot reflects the structure of a log-linear model assuming complete independence among the three categorical variables. Visual inspection of the shaded mosaic revealed deviations from expected frequencies under the independence model, particularly in the Cluster 3 × Treatment × Cycle combinations. Overrepresentation of 5-HT-treated sperm in Cluster 3 was detected at 10 cycles, whereas underrepresentation of vehicle-treated sperm in Cluster 3 was detected at 20 and 25 cycles. The residual patterns did not indicate substantial deviations in Clusters 1 or 2. These results suggest that the effect of treatment on sperm subpopulation distribution is not uniform across the cycles; rather, 5-HT stimulation appears to preferentially affect the dynamics of the subpopulation defined by Cluster 3 in a cycle-dependent manner.

## Cluster-specific mean values of motility parameters under normal and over-homogenized conditions

Exposure to 5-HT differentially and significantly affected the average values of motility parameters across clusters. Based on the patterns of velocity and linearity values, the clusters were named as follows: Cluster 1, "Low velocity–intermediate linearity"; Cluster 2, "High velocity–low linearity"; and Cluster 3, "Intermediate velocity–high linearity."

In conditions of normal homogenization, sperm in the distinct clusters respond differentially to 5-HT exposure. Sperm in cluster 1 modified the values of the motility parameters, except for LIN and BeatCross (Fig 6A). In cluster 1, velocities (VCL, VAP and VSL) were greater for those sperm exposed to 5-HT, but in cluster 2 velocities (VCL, and VAP) were

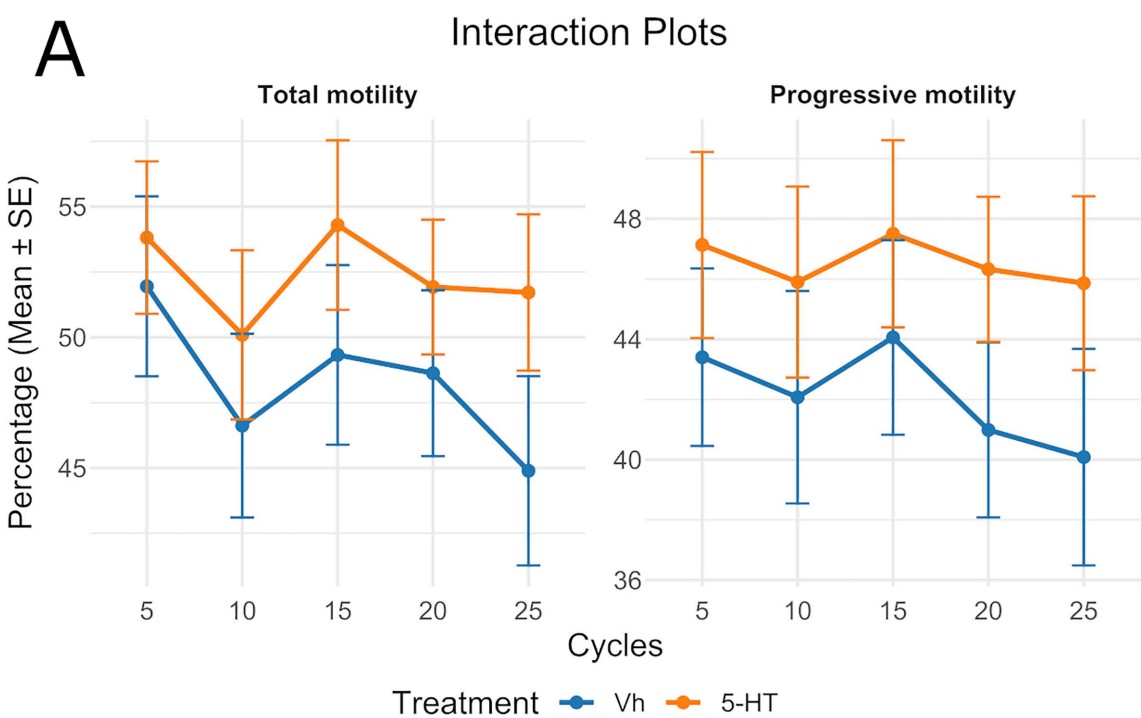

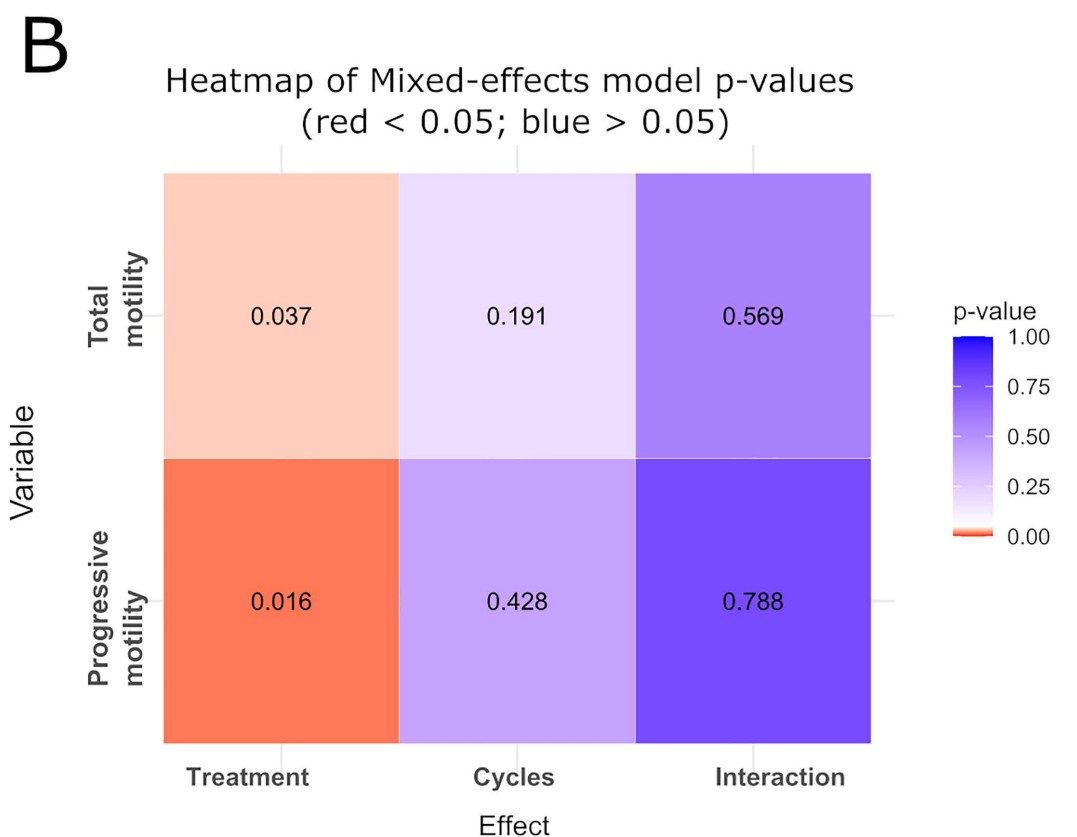

**Fig 2. Effects of 5-HT and homogenization cycles on sperm motility (A) Interaction plots for total and progressive motility.** Data are mean +/- standard error (N = 5). (B) Heatmap showing *p-values* for the effects of 5-HT, homogenization cycle and their interaction. The color of each square

indicates whether the p-value for treatment, number of cycles, or their interaction falls below or above the significance threshold of 0.05. Numbers inside each square represent the exact *p-value.*

lower or unchanged (VSL) (Fig 6C). In cluster 3, all motility parameters were modified by 5-HT, but VCL, STR, and ALH remained unchanged (Fig 6E).

In over-homogenized conditions, 5-HT, cycles, and their interaction significantly affected most motility parameters across all clusters (Fig 6B, D and F). For clusters 1 and 2, no clear pattern of 5-HT–induced changes was observed across cycles. In cluster 3, 5-HT exposure increased VCL, VAP, VSL, and ALH at intermediate cycle levels, but these parameters remained unchanged at higher cycle levels.

## Discussion

A device capable of objectively homogenizing semen samples was designed and constructed, with control over angle, velocity and number of cycles. The hardware and firmware of CASHo provide precision and repeatability, ensuring semen samples reach a state in which no further changes in sperm density occur. CASHo provides a purpose-built, open-hardware solution for the controlled homogenization of semen samples, addressing a need not covered by existing commercial or general-purpose laboratory devices. This assessment is based on technical information provided by manufacturers, as no direct experimental comparisons with commercial devices were performed in this study.

### Capabilities and limitations of CASHo

The designed CASHo possesses various advantages including (1) a highly accurate angle of rotation as a stepper motor was used. (2) The coefficient of variation in timing between cycles is very low, confirming high repeatability. (3) The angle and duration of rotation are highly precise. These enables objective and reproducible studies on semen homogenization by allowing precise control of rotation speed, angle, and number of cycles. Use of CASHo in experimental settings highlights the importance of identifying the point at which homogenization no longer alters sperm density, allowing the procedure to be stopped once an objective and repeatable state is reached.

In its current version, CASHo is an affordable homogenizer specifically designed for boar semen, but it can be readily adapted for use with semen from other species. The 50 mL tube holder can be replaced with holders of different capacities, such as 2 mL or 15 mL, depending on the experimental requirements. Likewise, the extruded aluminum base may be substituted with other readily available materials, including rectangular steel tubing or wood, which can be secured by welding or screwing. Furthermore, CASHo can be upgraded by modifying the firmware and incorporating a display and potentiometers to regulate parameters such as the number of routines, cycles, rotation angle, or speed.

### Limitations of existing commercial homogenization devices

In previous studies, the issue of semen homogenization has been addressed experimentally [21–25]; however, none of these studies provided an operational definition or specified a standardized protocol to homogenize semen samples. In some reports, homogenization was performed using equipment not designed for handling sperm samples, or equipment specifically designed for semen samples from a particular species [23,24]. This variation complicates replication and prevents the establishment of standardized homogenization protocols. Moreover, existing commercial equipment tested in prior reports [21–25] cannot be easily or adequately adapted to the requirement of different species.

Although no direct comparison with commercial homogenizers was conducted, our discussion of existing devices and their reported effects is based on previously published studies. Therefore, claims regarding their limitations should be interpreted in this context, and future work should include direct comparative testing to validate these observations experimentally. In the present study, the displacement velocity and rotation angle were determined empirically based

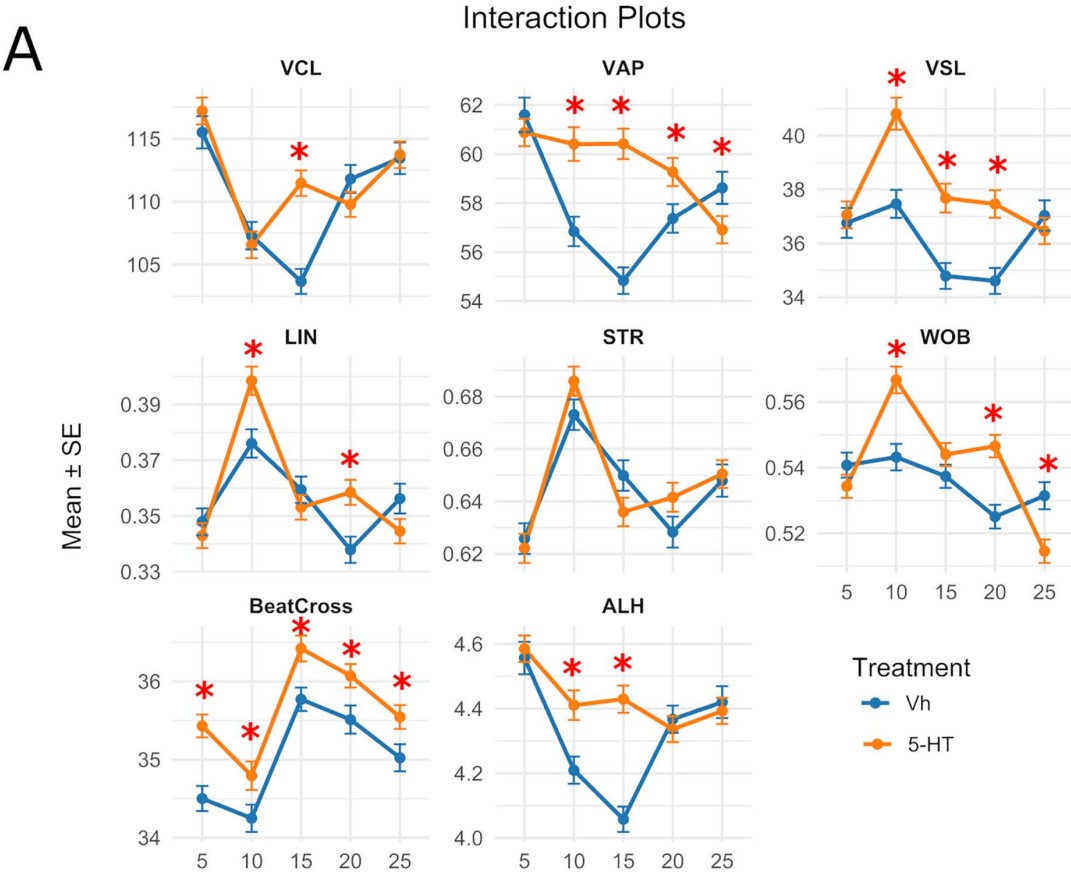

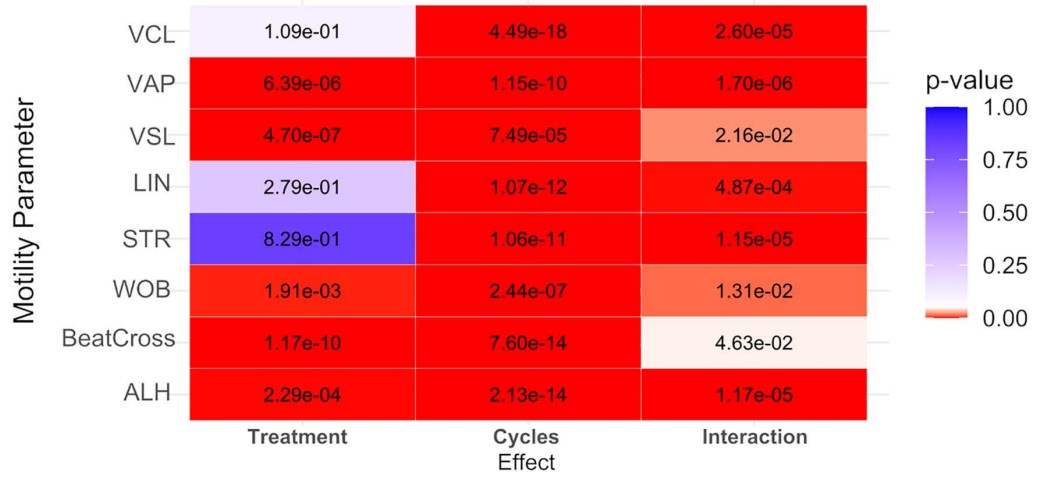

**Fig 3. The exposure to 5-HT and distinct number of cycles affected the values of motility parameters. (A)** Interaction plots showing average of kinematic parameters. Data are mean +/- standard error (N = 5). Asterisks indicate differences between Vh and 5-HT at each cycle level. **(B)** Heatmap showing p-values for the global effects. The color of each square indicates whether the p-value for treatment, number of cycles, or their interaction falls below or above the significance threshold of 0.05. Numbers inside each square represent the exact *p-value*.

**Table 2. Detail of Principal components analysis realized on motility parameters obtained from the computer assisted sperm analysis system.**

|  | PC1 | PC2 |
|---|---|---|
| Variance explained (eigenvalue) | 43.37 (3.47) | 35.34 (2.83) |
| **Eigenvectors *** |  |  |
| VCL | 4.24 | 26.78 |
| VAP | 17.10 | 12.39 |
| VSL | 25.54 | 0.25 |
| LIN | 15.91 | 14.56 |
| STR | 6.21 | 18.13 |
| WOB | 14.60 | 2.94 |
| BeatCross | 6.81 | 3.27 |
| ALH | 9.59 | 21.66 |

* Eigenvectors represent a degree of association with each one of motility descriptors.

VCL, curvilinear velocity; VAP, average path velocity; VSL, straight line velocity; LIN, linearity; STR, straightness; WOB, wobble; BeatCross, beat cross frequency; ALH, lateral head displacement.

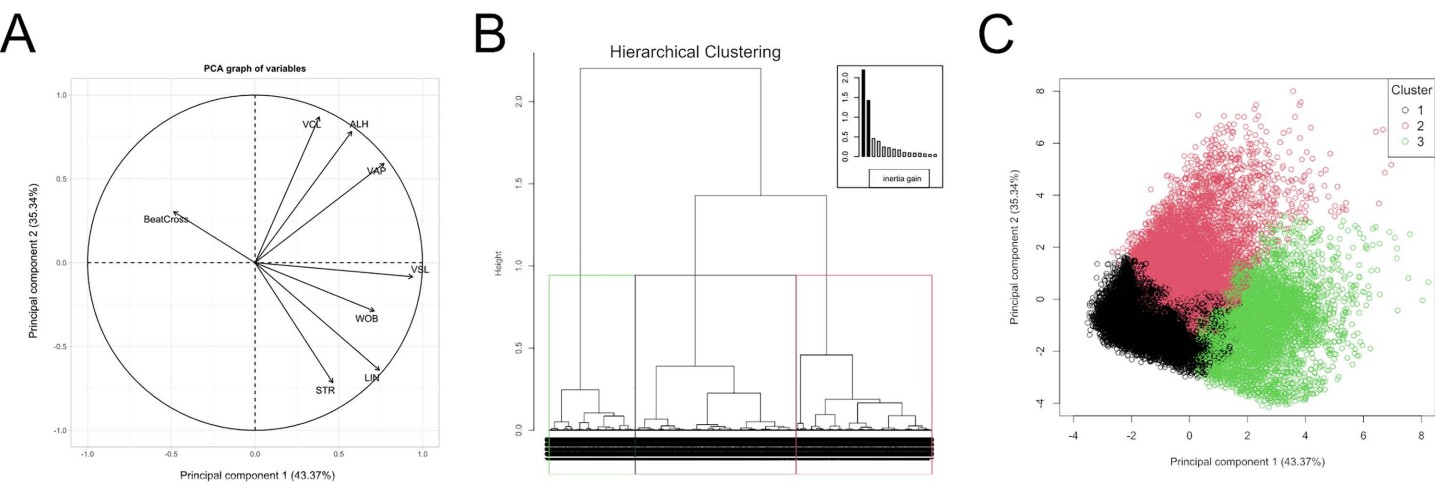

**Fig 4. Principal component analysis (PCA) was applied to reduce the dimensionality of the dataset, and hierarchical clustering was then used to identify kinematic subpopulations.** Data from 18,644 sperm, each characterized by eight motility parameters, served as input for the principal component analysis (dimensionality reduction step). **(A)** Correlations between motility parameters and Principal Components 1 and 2. Motility parameters are represented as vectors (arrows); therefore, the direction and length of each arrow indicate the sign and magnitude of the correlation. Data from the principal components were then used as input for the hierarchical clustering algorithm. **(B)** The number of clusters (branches into color rectangles) was determined based on the dendrogram and the inertia gain (inset bar plot). Inertia refers to variance, with total inertia (total variance) decomposed into between- and within-group components. The within-cluster inertia gain characterizes cluster homogeneity. Labels at the base of the dendrogram correspond to individual sperm IDs. **(C)** Scatterplot of the first two principal components showing clusters identified in **(B)**, color-coded according to cluster assignment.

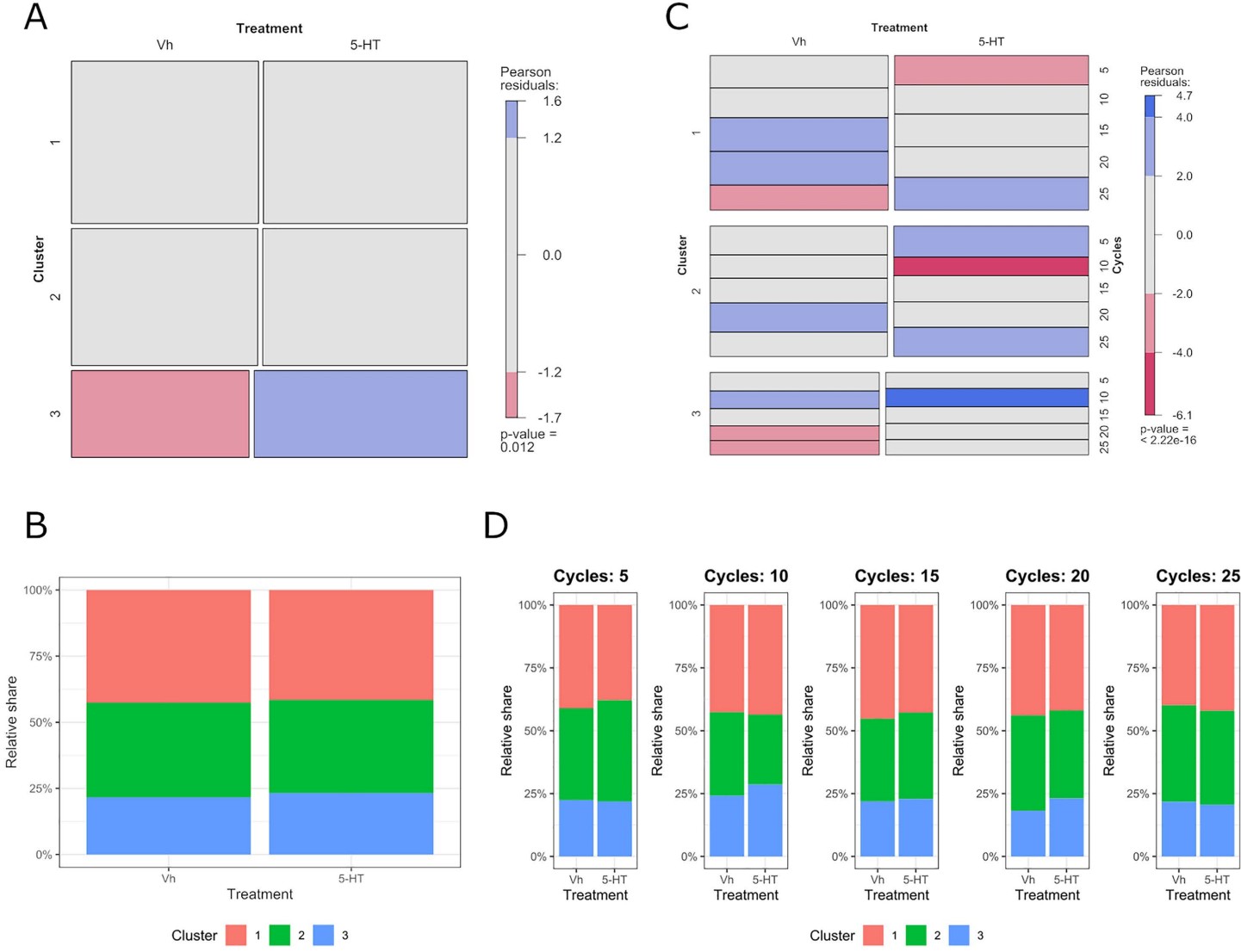

**Fig 5. Proportion of sperm within clusters in relation to the effects of 5-HT and homogenization cycles. (A)** Mosaic plot showing the proportion of sperm classified into each cluster (1, 2, and 3) for the control (Vh) and 5-HT treatment groups. The height of each tile represents the relative frequency of each treatment–cluster combination, while the width reflects the proportion of sperm assigned to each cluster. **(B)** Stacked plot showing the distribution of sperm percentages across clusters and treatments. **(C)** Mosaic plot showing distribution of sperm across treatments and homogenization cycles within each cluster. **(D)** Stacked plot showing the distribution of sperm percentages across clusters, cycles and treatments. For A and C, blue tiles indicate more sperm than expected under independence (positive residuals), red tiles indicate fewer sperm than expected (negative residuals), and grey tiles indicate values close to the expected distribution. The area of each tile represents the joint frequency of sperm within a given treatment–cycle–cluster combination. The color scale is shown in the lookup table to the right of the mosaic [19].

on hand-operated trials, before programming CASHo. Other researchers have examined the effect of manual versus mechanical homogenization on sperm motility; however, their study was conducted under conditions that are not directly comparable [21]. Sperm density, as well as total and progressive motility evaluated by CASA, were reported to be higher in diluted boar semen samples mixed with a vortex mechanical mixer (80% speed) compared to manual mixing (turning the tube approximately five times for 2–3 seconds). Moreover, homogeneity of the suspension in that study was assessed only visually [21]. The homogenization-like process has been described without specifying the approximate degrees of

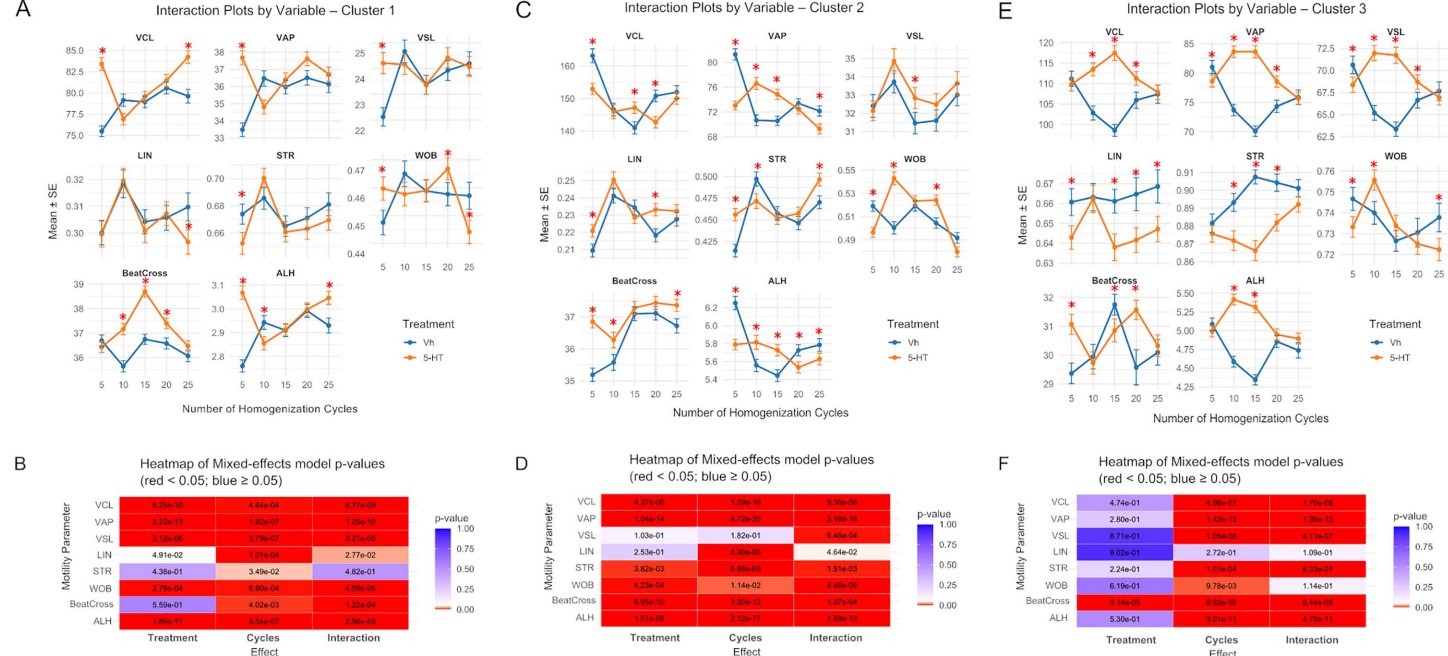

**Fig 6. Effects of 5-HT and homogenization cycles on sperm motility kinematic for each sub-population cluster.** (A, C and **E**) Interaction plots showing average of kinematic parameters of sperm classified into each cluster (1, 2, and 3) for the control (Vh) and 5-HT treatment groups. Asterisks in panels A, C, and E indicate significant differences between Vh and 5-HT treatments at each cycle level. Data are mean +/- standard error. (B, D and **F**) Heatmap showing *p-values* for the global effects into each cluster (1, 2, or 3); the color of each square indicates whether the p-value for treatment, number of cycles, or their interaction falls below or above the significance threshold of 0.05. Numbers inside each square represent the exact *p-value*.

inclination: agitated by gently rocking each tube five times once daily' [3]. When semen samples are processed in this manner, no effects on sperm motility kinematics were detected; however, manual rotation of 180° every 12 hours or five automatic 360° rotations per hour using a Kobimove Mini (Cobiporc, Saint Gilles, France) resulted in decreased boar sperm motility [25]. Similarly, equine sperm subjected to a rotation frequency of 5 revolutions per minute (rpm) and 36 rpm using a roller bench (model RM5, Minitüb GmbH) for four consecutive days showed reduced total motility [26].

In the present study, the serotonin experiment was conceived as a functional demonstration of CASHo's utility. CASHo was used to assess the effect of homogenization cycles on the motility parameters of sperm exposed to 5-HT. Two complementary analytical approaches were employed including evaluation of overall average values of sperm motility kinematics, and assessment within defined kinematic subpopulations.

## Average values of motility parameters versus subpopulation analysis

The low values of total and progressive motility measured by CASA in this study contrast with the threshold values established by CEIEEP. However, CASA systems generally report lower sperm motility compared to subjective evaluation by technicians [27]. With the CASA plugin used in this study, the main limiting factor reducing the number of motile sperm was the threshold set for the minimum number of tracks required to be classified as motile.

The analysis of average data showed no effects of 5-HT on sperm motility when semen samples were well-homogenized with five cycles. However, the clustering analysis allowed the identification of sperm motility kinematic subpopulations that responded differentially to 5-HT. Moreover, the motile behavior of these sperm subpopulations was differentially affected by 5-HT under over-homogenization conditions. The proportion of sperm in subpopulation 2 was over-represented at five

and 25 cycles but under-represented at 10 cycles, indicating a redistribution of sperm among subpopulations as a function of the homogenization conditions. Not all studies using CASA systems identify motility kinematics subpopulations; instead, many report only average values. It has been reported that sperm express proteins related to serotonergic communication, including receptors, transporters, and metabolizing enzymes [11]. In that study, stallion sperm were exposed to 5-HT under non-capacitating conditions, and motility was evaluated using CASA, but only average motility values were reported. Exposure to 5-HT increased sperm velocities (VCL, VAP, and VSL) [11]. Other studies have reported average motility values for sperm exposed to agonists or antagonists of 5-HT receptors under capacitating conditions [12,13].

In the present work, VCL and VAP values increased after exposure to 5-HT at five cycles in cluster 1, decreased in cluster 2, but showed no changes in cluster 3 or when the average values in the full dataset were analyzed. The fact that changes in values of motility parameters were evident at the subpopulation level indicates that relevant information is lost when only average values are considered in the full dataset. Previously, changes in the structure of kinematic subpopulations were detected in boar sperm exposed to ketanserin under non-capacitating conditions [14]. In that study, the subpopulation with the highest velocity values showed a significant increase in these parameters after exposure to ketanserin (Supplementary data in [14]), which support our results.

Sperm velocity is of biological importance, and a key determinant for fertility success [28]. In this study, the effects of 5-HT on sperm velocity varied among clusters. In the cluster with the highest VCL values at five cycles, all motility parameters except VSL showed significant differences. In that cluster, as the number of cycles increased (over-homogenization conditions), interactions with 5-HT effects became evident. These findings suggest that subpopulations have differential responses to 5-HT exposure, manifested as changes in kinematics. The molecular pathways and components underlying these 5-HT-induced changes in kinematics remain unknown. However, the existence of differentially responsive kinematic subpopulations may be linked to sperm development and maturation in the testis and epididymis, respectively [29], as well as to the factors involved and their level of expression. It is known that the $5\text{-HT}_2$ receptor stimulate the activation of phospholipase C and calcium signaling [30], and at average level the $5\text{-HT}_2A$ receptor has been associated with increased human sperm velocity [31].

### 5-HT and hyperactivated sperm motility

Capacitation is a key step in the fertilization process [32]; thus, extenders for long-term storage must prevent capacitation while maintaining motility. One event in the capacitation process is the acquisition of hyperactivated motility. Although serotonergic communication has shown to stimulate sperm hyperactivation [12,13], the 5-HT signaling for hyperactivation were not subjected to the present study. It has been shown that 5-HT stimulate $Ca^{2+}$ signaling, in vitro, to triggers sperm motility hyperactivation [32,33,35]. Semen extenders may contain chelating agents, such as EDTA, which have high affinity for $Ca^{2+}$ [34,35]; therefore, 5-HT-related hyperactivated motility is unlikely under the conditions of this study. Another factor required for hyperactivation is prolonged incubation [12,13,33], which is not included. Finally, the characteristic motility patterns described for hyperactivated sperm [12,35,36] were absent.

### Conclusions and future directions

CASHo is (1) a reliable open-hardware device for homogenization of semen samples. (2) An objective definition of homogenization of boar semen samples can be achieved using CASHo. (3) Exposure of sperm to 5-HT at five cycles differentially modified motility parameters among subpopulations, indicating that sensitivity of sperm to 5-HT varies across clusters. (4) 5-HT reduced motility parameter values in the subpopulation with highest VCL. (5) Over-homogenization introduced bias into kinematic results when sperm were exposed to 5-HT.

The effects of 5-HT and over-homogenization on sperm quality parameters other than motility remain unknown. Future research should investigate whether these factors influence sperm viability, membrane integrity, acrosome status, mitochondrial function, or DNA fragmentation, as their absence represents a limitation of the present study.

Because the number of cycles required to reach objective homogenization may vary across species, the threshold at which over-homogenization occurs is also likely to differ. Exploring these additional endpoints will help determine whether the effects detected on motility extend to other aspects of sperm physiology. Furthermore, studies across different species and extender formulations could clarify whether the impact of homogenization cycles and serotonergic signalling is generalizable or context-dependent.

## Supporting information

**S1 File. CASHo components, design files descriptions, build and operation instructions.**
(DOCX)

**S2 File. Dataset of duration of cycles.**
(CSV)

**S1 Video. Hand homogenization.**
(MP4)

**S2 Video. CASHo in operation.**
(MP4)

## Acknowledgments

The authors thank Martin Lavalle for his assistance in preparing isometric and technical draws of the STL models.

## Author contributions

**Conceptualization:** Andrés Aragón, María E. Ayala.

**Data curation:** Nancy A. Juárez-Contreras, Cindy U. Rivas-Arzaluz.

**Formal analysis:** Nancy A. Juárez-Contreras, Cindy U. Rivas-Arzaluz, María E. Ayala.

**Funding acquisition:** Andrés Aragón.

**Investigation:** Cindy U. Rivas-Arzaluz, María E. Ayala.

**Methodology:** Nancy A. Juárez-Contreras, Oscar Gutiérrez Pérez, María E. Ayala.

**Resources:** María E. Ayala.

**Supervision:** Andrés Aragón.

**Validation:** Nancy A. Juárez-Contreras, Cindy U. Rivas-Arzaluz.

**Writing – original draft:** Cindy U. Rivas-Arzaluz, María E. Ayala.

**Writing – review & editing:** Andrés Aragón.

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
