## [Decision Letter · Decision Letter 0]

21 Jul 2025

Dear Dr. Aragón,

Thank you for submitting your manuscript to PLOS ONE. After careful consideration, we feel that it has merit but does not fully meet PLOS ONE’s publication criteria as it currently stands. Therefore, we invite you to submit a revised version of the manuscript that addresses the points raised during the review process.

We look forward to receiving your revised manuscript.

Kind regards,

Sayyed Mohammad Hadi Alavi

Academic Editor

PLOS ONE

 [UNAM, DGAPA, PAPIIT, IN221018, IT201021 and IN224925.]. 

Additional Editor Comments:

Dear Prof. Andrés Aragón Martínez

Due date to submit revision: September 20, 2025

Thank you very much for your submission to PLOS ONE. I am glad to inform you that three reviewers with expertise in computer-assisted sperm analysis have reviewed your manuscripts, and provided us with very valuable comments that are appended below. They all highlighted novelty and originality of the present study, however there are many key concerns that require consideration upon revising the MS. Please let me describe in-brief the major comments, while I would appreciate your careful consideration to all of them when you revise the MS.

Some open-software for CASA has been previously introduced and published in the form of a “Research Article”. Under this category, the MS should follow the guidelines according to PLOS ONE. More information is available at:

https://journals.plos.org/plosone/s/what-we-publish

1) In particular, please briefly describe disadvantage and advantage of manual or automated homgenizer in Introduction, and open the idea for developing your device. Introduction in its current version misses a clear aim of your work. Moreover, as reviewer asked for, it is very important to include the rational behind using 5-HT to examine validity of your device. Hence 5-HT stimulates sperm hyper-motility, this could be a good way to examine whether sperm maintain its capacity for hyper-motility during the storage period.

2) Please open “Materials and Methods” next to “Introduction”, at which designs and construction of the hardware, operational instruction, technical issues and limitations, biological test for validation, and statistical analysis are placed.

We are very grateful to you for in-detail description of hardware; however several steps could be moved from the main body into the supplementary data, for instance the required materials and name of suppliers (Table 2).

The method for sperm collection, handling, and spermatozoa motility analysis could be properly described in M&M, as well as statistical analysis for comparison.

3) Open a result section at which you include the results of sperm density and motility

4) The MS needs “Discussion” in which the advantage of the hardware should be compared with existed manual or other type of homogenizers. In this section, the results of biological test could be described. In this latter case, please describe whether 5-HT stimulation of spermatozoa hyper-motility was maintained or any other physiological system that 5-HT acts on spermatozoa motility. Showing statistical difference is essential for Figure 10.

5) Please avoid redundant of results. I found data in Table 3 are identical to those shown in Figure 11.

6) Please reduce number of figures with making together some current figures. Another example is to move individual variation in sperm density (Figure 10) to supplementary data, and keep panel showing the mean values (bottom panel).

7) Please uniform “biological terms” through the MS, and edit the English before re-submission.

8) Please find examples of some open CASA software published previously, one is from PLoS Comput Biol:

Alquézar-Baeta C, Gimeno-Martos S, Miguel-Jiménez S, Santolaria P, Yániz J, Palacín I, et al. (2019) OpenCASA: A new open-source and scalable tool for sperm quality analysis. PLoS Comput Biol 15(1): e1006691. https://doi.org/10.1371/journal.pcbi.1006691

Butts, I.A., Ward, M.A., Litvak, M.K., Pitcher, T.E., Alavi, S.M., Trippel, E.A. and Rideout, R.M. (2011) Automated Sperm Head Morphology Analyzer for Open-Source Software. Theriogenology, 76, 1756-1761.

http://dx.doi.org/10.1016/j.theriogenology.2011.06.019

Wilson-Leedy JG, Ingermann RL, (2006) Development of a novel CASA system based on open source software for characterization of zebrafish sperm motility parameters, Theriogenology 67(3):661-72. https://doi.org/10.1016/j.theriogenology.2006.10.003

Once again, I appreciate choosing PLOS ONE for your work, and I hope our review provide you with valuable comments on your research.

With very best regards

Hadi Alavi

Reviewers' comments:

Reviewer's Responses to Questions

**Comments to the Author**

1. Is the manuscript technically sound, and do the data support the conclusions?

Reviewer #1: No

Reviewer #2: Partly

Reviewer #3: Yes

2. Has the statistical analysis been performed appropriately and rigorously?

Reviewer #1: Yes

Reviewer #2: Yes

Reviewer #3: Yes

3. Have the authors made all data underlying the findings in their manuscript fully available?

Reviewer #1: Yes

Reviewer #2: Yes

Reviewer #3: Yes

4. Is the manuscript presented in an intelligible fashion and written in standard English?

Reviewer #1: Yes

Reviewer #2: Yes

Reviewer #3: No

Reviewer #1: The manuscript “Welcome to the open-hardware sperm quality lab, featuring the new homogenizer: testing serotonin's effects on sperm motility” is aims to design, construct,

and validate an objective homogenizer for boar semen doses.

I appreciate the importance of studying the effects of semen homogenization in the laboratory, and at first glance, your manuscript may provide interesting data.

However, we have some concerns with the general contents and the presentation.

Major points

How many ejaculates from each boar were used in Experiment 1?

Semen collection records with lower sperm motility were excluded? What was the threshold criterion for total motility? The collection periods before and during the experiment did not account for the spermatic wave?

The authors analyze sperm kinematic variables and kinematic patterns in boar semen subpopulations. However, data on total and progressive motility (%) are not presented in the study. Since total and progressive motility are fundamental variables for semen analysis and the production of semen doses- whether fresh, chilled, or even frozen- the authors should include these results in the manuscript. This would help determine the effect of homogenization on these variables and address the current lack of evidence regarding basic motility parameters, as opposed to only advanced (kinematic) variables.

Can the authors provide data on the morphology of boar sperm? The authors could provide results on the effect of homogenization on sperm morphology, particularly regarding head and midpiece defects such as decapitated spermatozoa, in order to assess the impact of homogenization on the incidence of sperm morpho-anomalies. I agree with the authors that sperm quality is important to establish its relationship with male fertility; therefore, this information is essential for the study.

The experiment on serotonin is not clearly aligned with the central objective of the study (The goal of this study was to design, construct, and validate an objective homogenizer for boar semen doses). The authors should justify the inclusion of this experiment in the manuscript and explain how it relates—if at all—to the main objective of the article.

Finally, I believe this study provides valuable information regarding semen analysis methodology in the laboratory.

The largest issue with this study is that the experimental design is unclear. If it is the latter, then the differences observed may simply be differences between boars, and not between cycles. There is no model, and the authors do not provide one in their study.

Can the authors indicate a clear and summarized conclusion of this work?

Reviewer #2: Review Report

Title: Welcome to the open-hardware sperm quality lab, featuring the new homogenizer: testing serotonin's effects on sperm motility

Overall assessment

This manuscript presents a novel, open-source device (CASHo) for objective semen homogenization, and investigates its application in studies on serotonin’s effects on boar sperm motility. The technical innovation is commendable and valuable for labs with limited resources. However, the current structure of the manuscript overly emphasizes hardware details at the expense of biological analysis and interpretation.

The experimental design is of interest, and the identification of sperm kinematic subpopulations is a strong biological element. Yet the manuscript needs significant reorganization and revision to meet the scientific standards expected by PLOS ONE. Below are major and minor suggestions to improve clarity, structure, and scientific impact.

Major comments

Restructure the manuscript – move detailed hardware descriptions to supplementary materials

The current version devotes excessive space to construction details (e.g., STL file descriptions, wiring diagrams, firmware logic). This level of detail is valuable, but it should be moved to Supplementary Materials. The main manuscript should include only a concise overview highlighting why CASHo is different, how it improves over existing methods, and what the key design parameters are.

Retain calibration data as an experimental foundation

The experiment determining the number of cycles needed for full sperm homogenization is valuable and scientifically necessary. This should remain in the main manuscript, clearly defined as a preliminary calibration step for establishing the operational definition of “objective homogenization” prior to biological experiments. A brief rationale should be included to explain why 5 cycles was selected as the standard condition.

Elevate the biological relevance – center the subpopulation analysis

The most biologically meaningful part of the study is the identification of kinematic sperm subpopulations and their distinct responses to serotonin. This section should be expanded and moved into the main Methods and Results sections. Clearly explain the PCA and clustering strategy and provide justification for the choice of three clusters.

Analyze all subpopulations and whole sample

Only one subpopulation (Subpop 2) is presented in the main text. The rationale behind is not clear, nor, I think “biologically relevant”. Please include the full analysis for all three subpopulations and also show results for the entire unsegmented sperm population. This would allow readers to compare how much information is lost or gained by using subpopulation analysis versus pooled data.

Report the proportional distribution of subpopulations

It would strengthen the manuscript to include percentage representation of each cluster across all experimental groups (you have provided the numbers of sperm in each subpopulation, just add the percentage along with it). This could help detect whether serotonin or homogenization changes the relative size of clusters (e.g., shift from more motile to less motile populations), which is biologically significant.

Consider integrating additional sperm parameters.

If feasible, consider combining subpopulation data with other available sperm quality markers such as membrane integrity, mitochondrial activity, or acrosome status. This would deepen the biological understanding of each subpopulation and potentially reveal phenotype-function correlations. This could be addressed in additional experiments or simply highlighted as an important future direction—demonstrating the broader utility of both the CASHo device and subpopulation analysis, well beyond serotonin-related studies.

At line 444 you stated that subpopulation 2 is “biologically relevant”. I think this is not correct. Relevant to what? All sperm clusters were biological samples. Dividing them into three subclasses allows for a better understanding of the effect of serotonin—highlighting which subgroup is most sensitive to the treatment and which is most resistant. That’s biology: individual variation grouped into clusters or subpopulations. This approach is already interesting in itself, but it would be even more informative if the percentage of each subpopulation (from the most affected to the most resistant) was presented. Ideally, these data could be combined with other biological parameters of the sperm, such as membrane stability tests, mitochondrial status, etc. This would shed more light on the data and might allow for a higher-level interpretation of subpopulation characteristics—for example, by showing correlations between the proportion of spermatozoa with unstable membranes and the subpopulation most affected by serotonin (or perhaps another trend—I’m just speculating).

Clarify language and improve precision

The manuscript often uses vague or non-scientific terms. For example, Line 394 uses the word "observed" in a way that lacks analytical meaning. I recommend replacing it with a phrase such as “proven to be effective” or “shown to significantly alter 5 cycles”.

Minor comments

Abbreviations: Ensure all abbreviations (e.g., VCL, STR, etc.) are defined at first use.

Conclusion

This is a promising manuscript that combines bioengineering with reproductive biology, but a major revision is needed to shift focus from the technical construction to biological insight. The revised version should be more biologically oriented, clearly structured, and analytically precise.

Reviewer #3: The reviewed manuscript describes the development and validation of an open-hardware sperm homogenizer for boar semen, including a demonstration of its impact on sperm motility under several mixing regimes and after serotonin treatment. The topic is timely and relevant for laboratories working in reproductive biology and animal reproduction, especially in resource-limited settings. The authors present a high level of technical detail and reproducibility, including design files and firmware.

Nevertheless, the manuscript requires significant clarification, reframing, and improvement in presentation before it can be considered for publication.

Below are some major and minor comments for revision.

Major comments

Title of the manuscript:

The title “Welcome to the open-hardware sperm quality lab…” is catchy but somewhat misleading, as the manuscript primarily describes the development and validation of a single custom device, rather than a comprehensive laboratory platform. I recommend revising the title to focus on the sperm homogenizer (or mixer) development.

The title also places serotonin effects on equal level with the technical innovation. This is especially evident in the short title. However, the serotonin experiment appears to serve mainly as a demonstration of the device’s utility, rather than as a hypothesis-driven biological investigation. The serotonin aspect could likely be omitted from the title without loss of clarity.

Scientific justification for the serotonin experiment:

The rationale for combining serotonin treatment with over-homogenization is insufficiently substantiated. Why would over-homogenization interact with serotonin exposure in a way that demonstrates the value of the device? It is unclear whether this experiment was intended primarily as proof-of-concept or if it follows a mechanistic hypothesis. Please clarify the purpose and reframe the Results section if the goal is demonstration only.

Additionally, the term over-homogenization appears without justification. What exactly is meant by this term? How does it occur? What are its biological consequences? How can it be measured or prevented? Is it a species-specific phenomenon? These aspects should be discussed more explicitly.

Comparison with existing commercial devices:

The authors state that commercially available orbital or swing shakers cannot replicate manual hand motion and that this is important for semen handling. However, this claim lacks direct evidence or comparative testing. It would strengthen the study to cite published data or provide pilot comparisons showing, for example, that the motion replicated by the device results in better sperm distribution (e.g., lower variance in concentration or reduced damage to spermatozoa) compared to commercial mixers.

Terminology and clarity of processes:

While the manuscript consistently uses the term homogenization, it does not clearly distinguish the separate processes involved in semen handling. For clarity, I suggest distinguishing: 1. Re-suspension of spermatozoa (to redistribute settled cells); 2. Mixing of soluble additives such as serotonin; 3. Incubation time to allow interactions to occur.

Additionally, the term homogenization can imply mechanical disruption of cells, which may not be what the authors intend. Please clarify what exactly is meant in your usage of this term, and why homogenization is preferred over alternatives such as mixing or re-suspension.

The manuscript also uses word “sperm” inconsistently, sometimes referring to cells and sometimes to the fluid. Consider using spermatozoa when referring to the cells and semen when referring to the suspension.

Data presentation and statistical analysis:

While the manuscript provides a relatively detailed description of the statistical workflow, particularly for the motility subpopulation analysis, some key aspects remain unclear. Please clarify how biological replicates (i.e., semen samples from individual boars) were handled in the statistical models. If the same semen sample was split and subjected to multiple treatment conditions (e.g., different mixing cycles), this constitutes a repeated-measures design, and a repeated-measures ANOVA (or mixed model) may be more appropriate. If a standard ANOVA was used, please justify the assumption of independence and clarify whether boar ID was included as a blocking or random factor. Additionally, reporting effect sizes or confidence intervals alongside p-values would improve transparency and interpretability.

Language and style:

The manuscript contains numerous grammatical issues, inconsistent phrasing, and unclear terminology. For example, the phrase “homogenization of serotonin” is scientifically inaccurate. A thorough English language and grammar review is strongly recommended to improve clarity and readability, examples are given in minor comments.

Minor comments:

Line 28: “Correct number” — consider using “predictable number” or “consistent number” instead.

Line 39: “The effects of homogenization and overhomogenization…” — consider rephrasing as “potential over-homogenization.”

Line 45: The meaning of “Objective homogenization” is unclear

Line 53: Revise for clarity — suggest: “sperm motility is evaluated using a customized open-source computer-assisted sperm analysis system (CASA) [1].”

Line 54: “The evaluation of sperm concentration and vertical velocity is evaluated…” — rephrase to avoid redundancy.

Line 65: “Gentle homogenization is necessary because vigorous agitation can damage cells” — this is a key claim and should be supported with a reference.

Line 74: “Proprietary hardware is multi-purpose…” and “…specifically intended.”

Line 78: VAT is already a tax; “plus tax and VAT” is redundant.

Line 78: revise to “a programmable rotary agitator.”

Line 84: Clarify why “gentle” mixing is technically non-achievable with the referenced tool.

Line 86: Transition needed. For example: “One of the key applications of sperm homogenization is during sperm sample storage.” Or similar.

Line 87: “Commonly at 17°C” — is this is species-specific, I guess; please clarify.

Line 87: “Storage of samples leads to sedimentation of sperm cells, which must be homogenized before use.” Suggest rephrasing for clarity.

Lines 91–107: I suggest including references to positive consequences of resuspension of sperm during hypothermal storage reported e.g. in fish sperm.

Line 110: Why is the mentioned rotator not suitable for semen? Please elaborate.

Lines 122–124: The sentence should be rephrased for clarity.

Lines 136–137: Why should hand motion be replicated? Are there comparative studies on mixing efficiency or biological outcomes?

Lines 146–147: The device is said to work for “other cell types” — it would be more accurate to say “other cell suspensions,” and clarify that technical parameters can be adjusted, though effectiveness should be obviously tested case by case.

Line 354: Use “animal ID” instead of “boar.”

Line 358: Rephrase for clarity e.g., “Sperm concentration was higher in samples stored for less than 5 hours post-collection than after 24 hours.”

Line 382: Should be “We are interested in investigating…”

Line 382: Please clarify the rationale for testing serotonin in the context of device development.

Line 384: Define and justify the concept of “over-homogenization.” How is it measured? Is it species-specific?

Line 396: This step is more accurately described as “incubation with serotonin.” The term “homogenization of serotonin” is misleading.

Line 405: Note that 5 cycles were already shown in Experiment 1 to be sufficient for resuspension.

Line 414: Use “spermatozoa” instead of “sperm” when referring to individual cells.

Line 465: Why is an exact angle of rotation important in this context? Please justify.

Line 468: The claim that timing is precise should be supported with data; where is this tested?

Line 479: “The use of CASHo in experimental settings highlights the importance of achieving objective and repeatable homogenization without over-homogenizing.” : this was not tested nor discussed

Recommendation: Major Revision

The manuscript has high potential and presents a technically useful contribution. However, substantial revisions are needed to clarify terminology, justify experimental design, and align the structure and interpretation with the core goal of developing and validating a reproducible, low-cost sperm mixer-homogenizer.

**Do you want your identity to be public for this peer review?** For information about this choice, including consent withdrawal, please see our Privacy Policy

Reviewer #1: **Yes: ** Anthony Valverde

Reviewer #2: **Yes: ** Radosław Kowalski

Reviewer #3: No

---

## [Author Response · Author response to Decision Letter 1]

28 Aug 2025

The authors of the manuscript PONE-D-25-30420 “Welcome to the open-hardware sperm quality lab, featuring the new homogenizer: testing serotonin's effects on sperm motility” are grateful with Editor and reviewer's for their valuable suggestions to improve our manuscript. Each one of the comments and suggestions made by Editor and reviewers were addressed, as specified below. Please let me know any other comment or suggestion.

Best regards,

Dr. Andrés Aragón

PONE-D-25-30420

Welcome to the open-hardware sperm quality lab, featuring the new homogenizer: testing serotonin's effects on sperm motility

PLOS ONE

Dear Dr. Aragón,

Thank you for submitting your manuscript to PLOS ONE. After careful consideration, we feel that it has merit but does not fully meet PLOS ONE’s publication criteria as it currently stands. Therefore, we invite you to submit a revised version of the manuscript that addresses the points raised during the review process.

We look forward to receiving your revised manuscript.

Kind regards,

Sayyed Mohammad Hadi Alavi

Academic Editor

PLOS ONE

Answer: Thank you for your recommendations. The manuscript was fully re-structured. Style and file naming now follow the requirements of PloSOne.

[UNAM, DGAPA, PAPIIT, IN221018, IT201021 and IN224925.].

Answer: Funder had no role. The recommended sentence was included in the Cover Letter.

Answer: The recommended previously published works were revised and evaluated.

Additional Editor Comments:

Dear Prof. Andrés Aragón Martínez

Due date to submit revision: September 20, 2025

Thank you very much for your submission to PLOS ONE. I am glad to inform you that three reviewers with expertise in computer-assisted sperm analysis have reviewed your manuscripts, and provided us with very valuable comments that are appended below. They all highlighted novelty and originality of the present study, however there are many key concerns that require consideration upon revising the MS. Please let me describe in-brief the major comments, while I would appreciate your careful consideration to all of them when you revise the MS.

Some open-software for CASA has been previously introduced and published in the form of a “Research Article”. Under this category, the MS should follow the guidelines according to PLOS ONE. More information is available at:

https://journals.plos.org/plosone/s/what-we-publish

1) In particular, please briefly describe disadvantage and advantage of manual or automated homogenizer in Introduction, and open the idea for developing your device. Introduction in its current version misses a clear aim of your work. Moreover, as reviewer asked for, it is very important to include the rational behind using 5-HT to examine validity of your device. Hence 5-HT stimulates sperm hyper-motility, this could be a good way to examine whether sperm maintain its capacity for hyper-motility during the storage period.

Answer. Thank you for your valuable comments. The advantages and disadvantages of manual and automated homogenizers have now been included in the Introduction. The rationale for using 5-HT has also been incorporated. At the respect of hyperactivated motility stimulated by 5-HT we included the following text in Discussion “Capacitation is a key step in the fertilization process (Schmid et al., 2013); thus, long-term semen extenders must prevent capacitation while maintaining motility. Such is the case of the extender used in this study. An event in the capacitation process is the acquisition of hyperactivated motility. Serotonergic communication is related to hyperactivation (Fujinoki, 2011; Sakamoto et al., 2021); however, the in vitro development of hyperactivation requires compounds other than 5-HT, such as Ca²⁺, and in the case of boar sperm, additional compounds are needed (Harayama, 2018). Semen extenders may incorporate chelating agents, such as EDTA, which have high affinity for Ca²⁺ (Johnson et al. 2000; Gadea, 2003); thus, hyperactivated motility related to exposure to 5-HT is unlikely. Another factor required to achieve hyperactivation is prolonged incubation (Fujinoki, 2011; Harayama, 2018, Sakamoto et al., 2021), a condition not included in our study. Finally, characteristic motility patterns described for hyperactivated sperm (Suarez, 2003, Fujinoki, 2011; Harayama, 2018) were not observed in our study”.

Gadea J. Review: semen extenders used in the artificial inseminarton of swine. Spanish Journal of Agricultural Research. 2003 June 1;1(2):17–27.

Fujinoki M. Serotonin-enhanced hyperactivation of hamster sperm. Reproduction. 2011;142(2):255–66.

Harayama H. Flagellar hyperactivation of bull and boar spermatozoa. Reprod Med Biol. 2018;17(4):442–8.

Johnson LA, Weitze KF, Fiser P, Maxwell WM. Storage of boar semen. Anim Reprod Sci. 2000;62(1–3):143–72. Gadea J. Review: semen extenders used in the artificial inseminarion of swine. Spanish Journal of Agricultural Research. 2003 June 1;1(2):17–27.

Sakamoto C, Fujinoki M, Kitazawa M, Obayashi S. Serotonergic signals enhanced hamster sperm hyperactivation. J Reprod Dev. 2021;67(4):241–50.

2) Please open “Materials and Methods” next to “Introduction”, at which designs and construction of the hardware, operational instruction, technical issues and limitations, biological test for validation, and statistical analysis are placed.

We are very grateful to you for in-detail description of hardware; however several steps could be moved from the main body into the supplementary data, for instance the required materials and name of suppliers (Table 2).

The method for sperm collection, handling, and spermatozoa motility analysis could be properly described in M&M, as well as statistical analysis for comparison.

Answer. Thank you for your recommendations. We have implemented the following changes accordingly:

1) The “Materials and Methods” section has been relocated immediately after the “Introduction”;

2) Table 2, which included the list of required materials and suppliers, has been moved to the supplementary repository, along with the CASHo design files;

3) The procedures for sperm collection, handling, and spermatozoa motility analysis have been relocated and described in detail within the “Materials and Methods” section.

3) Open a result section at which you include the results of sperm density and motility

Answer. Thank you for your recommendation. A section with results of sperm concentration and motility was added to the Results.

4) The MS needs “Discussion” in which the advantage of the hardware should be compared with existed manual or other type of homogenizers. In this section, the results of biological test could be described. In this latter case, please describe whether 5-HT stimulation of spermatozoa hyper-motility was maintained or any other physiological system that 5-HT acts on spermatozoa motility. Showing statistical difference is essential for Figure 10.

Answer. Thank you for your comments. A Discussion section was added; in this section advantage of CASHo over devices available in the market and open-source are discussed. In the Discussion section the results of stimulation of sperm with 5-HT are also discussed. Factors needed for activated motility are mentioned. In Figure 10 (now Figure 1) the significant changes are shown.

5) Please avoid redundant of results. I found data in Table 3 are identical to those shown in Figure 11.

Answer. Thank you for your observation. Table 3 was deleted, and a new figure was added to show the proportions of sperm in each subpopulation and treatment (as reviewer 2 requested). Significant differences are included in the text and in heatmaps.

6) Please reduce number of figures with making together some current figures. Another example is to move individual variation in sperm density (Figure 10) to supplementary data, and keep panel showing the mean values (bottom panel).

Answer. Thank you for your valuable comment. We move Figure 1-9 to a document called CASHo_supplementary_details.pdf, which was moved to the repository https://osf.io/esfwg/, at the Supplementary information section.

7) Please uniform “biological terms” through the MS, and edit the English before re-submission.

Answer. Thank you for your recommendation. The “biological terms” have been carefully revised and standardized throughout the manuscript. In addition, the English language has been edited by a professional service with native English speakers.

8) Please find examples of some open CASA software published previously, one is from PLoS Comput Biol:

Alquézar-Baeta C, Gimeno-Martos S, Miguel-Jiménez S, Santolaria P, Yániz J, Palacín I, et al. (2019) OpenCASA: A new open-source and scalable tool for sperm quality analysis. PLoS Comput Biol 15(1): e1006691. https://doi.org/10.1371/journal.pcbi.1006691

Butts, I.A., Ward, M.A., Litvak, M.K., Pitcher, T.E., Alavi, S.M., Trippel, E.A. and Rideout, R.M. (2011) Automated Sperm Head Morphology Analyzer for Open-Source Software. Theriogenology, 76, 1756-1761.

http://dx.doi.org/10.1016/j.theriogenology.2011.06.019

Wilson-Leedy JG, Ingermann RL, (2006) Development of a novel CASA system based on open source software for characterization of zebrafish sperm motility parameters, Theriogenology 67(3):661-72. https://doi.org/10.1016/j.theriogenology.2006.10.003

Answer. Thank you for providing examples of open-source CASA software. Our manuscript does not describe the development of a new CASA software. Instead, it reports the use of an existing open-source plugin (Rivas et al., 2022), which is based on previously published tools. These foundational publications were properly cited in Rivas et al., (2022): “We looked at three to four fields per sample and analysed the image sequences using ImageJ software version 1.50d (Rasband, 1997) and the plugin developed by Wilson-Leedy and Ingermann (2007), later modified by Giaretta et al. (2017). We further adapted the plugin (CASA_bgm) for boar sperm by adjusting the parameter settings.”

Giaretta, E., Munerato, M., Yeste, M., Galeati, G., Spinaci, M., Tamanini, C., Mari, G., & Bucci, D. (2017). Implementing an open-access CASA software for the assessment of stallion sperm motility: Relationship with other sperm quality parameters. Animal Reproduction Science, 176, 11–19. https://doi.org/10.1016/j.anireprosci.2016.11.003

Rivas, C. U., Ayala, M. E., & Aragón, A. (2022). Effect of various pH levels on the sperm kinematic parameters of boars. South African Journal of Animal Science, 52(5), 693–704. https://doi.org/http://dx.doi.org/10.4314/sajas.v52i5.13

Wilson-Leedy JG, Ingermann RL, (2006) Development of a novel CASA system based on open source software for characterization of zebrafish sperm motility parameters, Theriogenology 67(3):661-72. https://doi.org/10.1016/j.theriogenology.2006.10.003

Once again, I appreciate choosing PLOS ONE for your work, and I hope our review provide you with valuable comments on your research.

With very best regards

Hadi Alavi

Reviewer #1:

The manuscript “Welcome to the open-hardware sperm quality lab, featuring the new homogenizer: testing serotonin's effects on sperm motility” is aims to design, construct, and validate an objective homogenizer for boar semen doses.

I appreciate the importance of studying the effects of semen homogenization in the laboratory, and at first glance, your manuscript may provide interesting data.

However, we have some concerns with the general contents and the presentation.

Major points

How many ejaculates from each boar were used in Experiment 1?

Answer. Tank you for your inquiry. For the experiment evaluating sperm concentration at different number of cycles, one semen sample from each of five boars was used. Five aliquots were taken from each semen sample. The five aliquots from each sample were individually processed using 5, 10, 15, 20, or 25 homogenization cycles with CASHo, one aliquot per cycle level. This information was added to the manuscript.

Semen collection records with lower sperm motility were excluded? What was the threshold criterion for total motility? The collection periods before and during the experiment did not account for the spermatic wave?

Answer. Thank you for your inquiries. The five boars used in this study were semen donors from the Liquid Gene Transfer Center (LGTC) at the CEIEPP, Faculty of Veterinary Medicine and Animal Science, UNAM. These animals were selected based on proven fertility and are routinely used in genetic dissemination programs for regional pig producers, as well as for pig production at the center’s experimental farm.

All donor boars follow a strict management protocol, which includes weekly semen collections. This schedule ensures consistent semen quality in terms of volume and concentration. Collections with progressive motility below 80% or with more than 15% morphological abnormalities are routinely discarded. Therefore, only ejaculates meeting these quality criteria were used in the present experiment. Samples were subsequently transported and evaluated before and after dilution using a commercial extender (1:1 volume-to-volume), as described in the manuscript. This information was added to the Material and Methods.

---

## [Decision Letter · Decision Letter 1]

2 Oct 2025

Dear Dr. Aragón,

Thank you for submitting your manuscript to PLOS ONE. After careful consideration, we feel that it has merit but does not fully meet PLOS ONE’s publication criteria as it currently stands. Therefore, we invite you to submit a revised version of the manuscript that addresses the points raised during the review process.

We look forward to receiving your revised manuscript.

Kind regards,

Sayyed Mohammad Hadi Alavi

Academic Editor

PLOS ONE

Journal Requirements:

Additional Editor Comments:

Dear Prof. Andrés Aragón

Thank you very much for revising the MS. Referees kindly accepted to review the MS-R1. We appreciate your efforts to revise the MS, however it still needs a "Major Revision". Please kindly let me ask you another revision that needs your careful consideration to the comments appended below. In particular, structuring, analyses of the data, presentation of results, and English of the MS needs revision.

Very best regards

Hadi Alavi

AE, PLOS ONE

Reviewers' comments:

Reviewer's Responses to Questions

**Comments to the Author**

Reviewer #1: (No Response)

Reviewer #2: (No Response)

Reviewer #3: (No Response)

2. Is the manuscript technically sound, and do the data support the conclusions?

Reviewer #1: Partly

Reviewer #2: Partly

Reviewer #3: Yes

3. Has the statistical analysis been performed appropriately and rigorously?

Reviewer #1: No

Reviewer #2: Yes

Reviewer #3: Yes

4. Have the authors made all data underlying the findings in their manuscript fully available?

Reviewer #1: Yes

Reviewer #2: No

Reviewer #3: Yes

5. Is the manuscript presented in an intelligible fashion and written in standard English?

Reviewer #1: Yes

Reviewer #2: Yes

Reviewer #3: Yes

Reviewer #1: The manuscript “Welcome to the open-hardware sperm quality lab, featuring the new homogenizer: testing serotonin's effects on sperm motility” is aims to design, construct, and validate an objective homogenizer for boar semen doses.

The revision substantially addresses the core reviewer concerns (design, modelling, structure, subpopulation analysis, clarity). The methodology is largely sound for a research article, with two moderate limitations: (i) absent morphology/functional markers and (ii) no empirical benchmarking vs. standard mixers. The statistical framework is appropriate overall, but I recommend (a) pairing/mixed modelling also at 5-cycle comparisons and (b) stating multiplicity handling or emphasizing effect sizes/CIs before acceptance.

I will recommend a moderate revision (address stats presentation at 5 cycles; add a brief multiplicity statement; explicitly flag morphology/functional markers as a limitation in Discussion; consider tightening the title). If these are implemented, I’d consider it publishable on the strength of its open-hardware contribution and transparent methodology

Reviewer #2: Manuscript title: Welcome to the open-hardware sperm quality lab, featuring the new homogenizer: testing serotonin's effects on sperm motility

Manuscript ID: PONE-D-25-30420_R1

Overall assessment

This manuscript presents the development and validation of an open-hardware semen homogenizer (CASHo) and its application in evaluating the effects of serotonin on boar sperm motility. The technical innovation is commendable and valuable for resource-limited laboratories. The operational definition of homogenization and the open availability of design files are important contributions.

However, the current manuscript in my opinion still requires major revisions. The main issues concern clarity of objectives, framing of the biological experiments, statistical transparency, and overemphasis on technical details. With revisions, this study could provide a meaningful resource for both bioengineering and reproductive biology.

Major comments

1. Clarity of objectives

The manuscript presents two parallel goals: (1) developing and validating the CASHo device, and (2) testing serotonin’s effects on sperm motility. At present, the serotonin experiment is insufficiently justified and feels appended.

Please clarify in the Introduction and Discussion whether the serotonin study is intended as a proof-of-concept demonstration of the device’s utility, or as a hypothesis-driven biological experiment. The framing will affect how results are interpreted.

2. Title and scope

The title (“Welcome to the open-hardware sperm quality lab…”) is catchy but misleading, since the work focuses on one device, not a full laboratory platform.

Consider a more precise title highlighting the homogenizer itself (e.g., “Development of an open-hardware semen homogenizer and application to serotonin effects on sperm motility”).

3. Biological relevance and missing data

While motility subpopulation analysis is valuable, basic semen quality measures (total and progressive motility, morphology, integrity) should be reported alongside advanced kinematic parameters.

The absence of sperm morphology or integrity data limits assessment of the device’s impact on sperm quality. At minimum, please acknowledge this limitation more directly in the Discussion.

4. Biological interpretation of clustering

While subpopulation analysis of sperm motility is an interesting approach, the manuscript does not fully explain what additional biological insights this provides compared to conventional mean values. For instance, the percentage shift (relative share) between subpopulations under different regimes and treatments should be visualized and discussed. This would clarify whether clustering reveals meaningful biological patterns (e.g., redistribution of sperm between more and less motile groups) beyond what is observed from average parameters.

At present, the added value of clustering remains unclear. Please expand the Discussion to explain what new information clustering contributes, how it should be interpreted biologically, and why it matters compared to traditional analyses.

5. Comparisons with existing methods

The manuscript claims that commercial devices are unsuitable for gentle homogenization, but no direct experimental comparison is presented.

Please either:

a) provide direct comparative data (preferred), or

b) clearly state that claims are based on published studies and acknowledge the lack of head-to-head testing.

6. Terminology and conceptual framing

The term homogenization is potentially misleading, as it usually refers to mechanical disruption of cells. Please clarify early in the text that the intended meaning is resuspension/mixing of spermatozoa and solutes.

Distinguish between different processes: (1) re-suspension of spermatozoa after sedimentation, (2) mixing of soluble additives such as serotonin, and (3) incubation for biological effects.

7. Over-homogenization

The concept of “over-homogenization” is central but underdeveloped. Please define it operationally (e.g., cycles beyond stabilization of sperm concentration) and explain its biological consequences.

Clarify whether this is species-specific (boar semen) or a generalizable phenomenon.

8. Discussion writting

In several places, the Discussion begins with references to other studies rather than with the authors’ own findings. It is generally clearer to start each paragraph by restating the relevant result from the current study, and then relate it to existing literature. For example, in lines 557–565, the text does not clearly connect the study’s own data to the cited work. I recommend restructuring the Discussion so that the authors’ results are presented first, followed by comparison with previous findings (whether in agreement, contradiction, or expansion), along with possible explanations for similarities or differences. This will strengthen the logical flow and emphasize the study’s contribution.

Minor comments

Organization and length

The revised structure (Methods following Introduction, supplementary hardware details moved to OSF) is much better. However, the main text is still too long (wordy). Consider further condensing hardware descriptions and focusing more on biological findings.

1. Figures and tables

Some figures remain redundant or overly detailed. For example, raw individual variability plots may be better suited as Supplementary Materials, with main text showing summarized results.

In addition to reporting kinematic parameters, the manuscript could benefit from a visualization of how the relative proportion of sperm subpopulations shifts under different homogenization regimes and treatments. Presenting percentage shares (e.g., stacked bar plots, mosaic plots, or similar) would make these shifts more intuitive and highlight whether serotonin or over-homogenization leads to redistribution between motile and less motile clusters.

2. Language and style

Although professionally edited, some vague terms remain. For example:

“Objective homogenization” → clarify with operational definition at first mention.

Replace observational wording (“we observed”) with more precise analytical phrasing where appropriate.

Ensure consistent use of spermatozoa (cells) versus semen (ejaculate fluid).

Recommendation

I recommend major revision. The technical contribution is strong and the study has real potential, but the biological framing, clarity of objectives, and statistical transparency must be improved before publication.

Reviewer #3: I want to thank the authors for the work done. The revision addresses several concerns raised by the reviewers. Methods are clearer, the statistics read more smoothly, figures and tables are better organized, and the rationale for including serotonin (5-HT) has more substance than before. Thank you for the careful work. To help the biology stand alongside the engineering, and so that the paper reads as a full study rather than a device demonstration, adding a few reader-oriented clarifications would make a noticeable difference. Below are my major and minor comments:

The biological narrative is almost there but still feels under-explained in places. In the Results, it would help readers if the motility clusters were briefly characterized in biological terms and not only statistically. A small table or inset with centroids (mean ± SE for the standard CASA variables) and a short descriptive label for each cluster would anchor the interpretation. Alongside that, the text and figure captions would benefit from one or two plain sentences that say, in straightforward language, how VCL, VAP and VSL change across cycles in vehicle versus 5-HT, including any non-monotonic patterns. Naming the reference condition explicitly (for example, vehicle at 5 cycles) will keep the reader oriented. These are small additions, but they will make the biology feel complete and not merely illustrative.

A related point concerns the captions themselves. Many readers, especially reading online (which is exactly the case of PLOS), arrive at the figures first and may rely on them heavily. It would be worth expanding the captions so they stand on their own: state what the plot shows, what comparison or test is displayed, the reference level used for contrasts, and the sample size for each panel. For the mosaic or heatmap-style figures, a single sentence on “how to read this” (e.g., what positive/negative residuals mean) will lower the barrier for speedy comprehension.

On the presentation of the device, a few “features” are currently placed in the Methods section as if they had been empirically established, although most are not demonstrated in the present study. Those statements would sit more comfortably in the device description as design intent or potential applicability, while the Results remain strictly limited to what was tested in boar semen under the reported conditions. This keeps the tone conservative without dampening the ambition of the tool (which was probably the intent of the authors, but not good for proper scientific paper).

Title choice is, of course, the authors’ decision. From a reader’s perspective, the current wording still suggests that the serotonin component shares equal billing with the device. A slightly more device-forward title would set expectations cleanly, while the strengthened biological explanations above would preserve the balance inside the paper. If the title is retained, ensuring that the Results and Discussion deliver the clear biological takeaways sketched above should address the potential mismatch.

The transition in the Introduction from device motivation to the serotonin example remains a little abrupt. A neutral linking sentence around the lines 118-119 would resolve this. For instance: “Proper re-suspension of semen is particularly critical when evaluating the effects of exogenous agents on spermatozoa function (e.g., responses to serotonin), because uneven mixing can confound exposure and bias motility readouts.”

Language and terminology are much improved but a few edits remain (see the Line comments below). A light copy-edit will catch the residual duplications and errors. On statistics, the current description is clearer; if feasible, adding effect sizes or confidence intervals for the primary outcomes and a brief note that assumptions were checked would enhance transparency. One sentence that states the experimental unit for each analysis, and how biological versus technical replication was handled, would also help readers follow the modeling choices. Finally, if precision of angle or timing is presented as an empirical outcome rather than a design specification, pointing to the relevant QA will close that loop.

In sum, only modest refinements are needed: a clearer biological “voice” for the clusters and velocity patterns, fuller captions that work for web readers,and conservative placement of undemonstrated device features. With these in place, the manuscript will read as a balanced, full-value study that showcases the homogenizer and what it enables

Line 107: “Given this lack of standardization, one consistent and fundamental aspect of homogenization is the manual movement of the tube.” - consider “remains the manual movement of the tube.”

Lines 114–115:“Despite this lack of standardization, one consistent and fundamental aspect is the manual movement of the tube.” - duplicates Line 107; please remove.

Line 116: “The design…” (capital “T” appears lost).

Line 133: “additional homogenisation” - this is a good place to introduce and define “over-homogenization”; you use the term later in the aims.

Line 167: “device” (minor typo).

Lines 283–284: Please rephrase for clarity: “At this stage, the semen samples were homogenized, but incubation with 5-HT was still ongoing. The samples were then mixed for 5, 10, 15, 20, or 25 cycles.”

Line 375: “After 5 to 25 cycles, the sperm concentration stabilized, reaching …” - clarify after how many cycles stabilization occurs (e.g., “stabilized by 5 cycles”).

Line 395: “five cycles at 4.9 ± 2.9 seconds with an inclination angle of 180 degrees” - it is unclear how the cycle duration was chosen; please state the criterion/rationale.

Line 396: “motility. At this stage, the semen sample was homogenized, but incubation with serotonin was still ongoing.” - please add where/when 5-HT was added relative to baseline mixing and subsequent cycles.

Line 415: “Sperm exposed to 5-HT exhibited reduced total and progressive motility” - compared to what?

Line 416: “Neither the effect of cycles nor the interaction between treatment and cycles was significant” - in which group/analysis, please specify.

Line 417: “However, significant effects of 5-HT, cycles, and their interaction were observed” - for which comparisons (e.g., at >15 or 25 cycles?)-please specify.

Line 495: “Sperm in Cluster 3 were less responsive to 5-HT treatment but remained sensitive to …” - do you mean no significant 5-HT effect at the initial step before additional cycles? Please clarify the intended comparison.

Line 506 - “Capabilities and limitations” - some items were tested here, others not; please separate demonstrated vs prospective capabilities (or cite prior validation if applicable).

Line 513 - “CASHo enables objective studies on semen homogenization.” - too general; consider a more concrete wording tied to the operational definition and reproducible cycles.

Lines 514–515 - “The 50 mL tube holder can be replaced with holders of other capacities (e.g., 2 mL or 15 mL).” - Were these tested? If not, frame as potential applicability; if yes, state evidence that mixing is adequate in smaller tubes.

Lines 529-530: “commercial equipment cannot be easily …” - consider “existing commercial equipment tested in prior reports cannot be easily …” (and good to cite the specific sources).

Line 586:“thus, long-term semen extenders” - consider “extenders for long-term storage.”

**Do you want your identity to be public for this peer review?** For information about this choice, including consent withdrawal, please see our Privacy Policy

Reviewer #1: **Yes: ** Anthony Valverde

Reviewer #2: **Yes: ** Radoslaw Kowalski

Reviewer #3: **Yes: ** Vitaliy Kholodnyy

---

## [Author Response · Author response to Decision Letter 2]

19 Oct 2025

The authors of the manuscript PONE-D-25-30420 “Welcome to the open-hardware sperm quality lab, featuring the new homogenizer: testing serotonin's effects on sperm motility” are grateful with Editor and reviewer's for their valuable time and suggestions to improve our manuscript. Each one of the comments and suggestions made by Editor and reviewers were addressed, as specified below. Please let me know any other comment or suggestion.

Best regards,

Dr. Andrés Aragón

Reviewer #1: The manuscript “Welcome to the open-hardware sperm quality lab, featuring the new homogenizer: testing serotonin's effects on sperm motility” is aims to design, construct, and validate an objective homogenizer for boar semen doses.

The revision substantially addresses the core reviewer concerns (design, modelling, structure, subpopulation analysis, clarity). The methodology is largely sound for a research article, with two moderate limitations: (i) absent morphology/functional markers and (ii) no empirical benchmarking vs. standard mixers. The statistical framework is appropriate overall, but I recommend (a) pairing/mixed modelling also at 5-cycle comparisons and (b) stating multiplicity handling or emphasizing effect sizes/CIs before acceptance.

I will recommend a moderate revision (address stats presentation at 5 cycles; add a brief multiplicity statement; explicitly flag morphology/functional markers as a limitation in Discussion; consider tightening the title). If these are implemented, I’d consider it publishable on the strength of its open-hardware contribution and transparent methodology

Answer: Thank you for your kind comments. We are grateful for your time and effort to help us to improve our manuscript. We hope that our responses and the corresponding modifications meet your expectations:

(i) We have added a statement in the Discussion explicitly acknowledging the absence of sperm morphology and integrity data as a limitation of the study. The text reads “Future research should investigate whether these factors influence sperm viability, membrane integrity, acrosome status, mitochondrial function, or DNA fragmentation, as their absence represents a limitation of the present study.”

(ii) We have added a statement in first paragraph of Discussion explicitly acknowledging the lacking of empirical benchmarking vs. standard mixers data. The text reads “CASHo represents a valuable alternative to commercial or open-hardware devices not specifically intended for semen homogenization. This assessment is based on technical information provided by manufacturers, as no direct experimental comparisons with commercial devices were performed in this study.”

(a) In the Statistics section, we originally wrote: “The effect of 5-HT at five cycles on total motility, progressive motility, and kinematic parameters (VCL, VAP, VSL, LIN, STR, WOB, BeatCross, and ALH) was compared using Student’s t-test,” because five cycles were considered the condition at which the sample was fully homogenized. Under this well-homogenized condition, we compared the Vh and 5-HT groups for total motility, progressive motility, and kinematic parameters. Subsequently, the five-cycle condition was used as the reference level in the mixed-effects model to analyze the over-homogenization conditions. To clarify we rephrased as follows “At five cycles, the effect of 5-HT treatment on total motility, progressive motility, and kinematic parameters (VCL, VAP, VSL, LIN, STR, WOB, BeatCross, and ALH) was assessed separately using a Student’s t-test, since this condition represents the point of complete homogenization and the comparison focused solely on treatment (5-HT vs. vehicle). Subsequently, the five-cycle condition was used as the reference level in the mixed-effects model, fitted to account for potential variability between boars, to evaluate over-homogenization effects across 10, 15, 20, and 25 cycles.”

(b) We thank the reviewer for this valuable suggestion. Because the aim of this study was exploratory rather than confirmatory, no formal multiplicity correction was applied. Instead, we placed emphasis on the magnitude and precision of the estimated effects, which are reported as effect sizes and 95% confidence intervals for fixed effects and interactions in S2 Table. This approach allows readers to evaluate the strength and direction of the observed effects beyond statistical significance.

Finally, the title was modified accordingly; it now reads “Development of an open-hardware semen homogenizer and application to serotonin effects on sperm motility.”

Reviewer #2: Manuscript title: Welcome to the open-hardware sperm quality lab, featuring the new homogenizer: testing serotonin's effects on sperm motility

Manuscript ID: PONE-D-25-30420_R1

Overall assessment

This manuscript presents the development and validation of an open-hardware semen homogenizer (CASHo) and its application in evaluating the effects of serotonin on boar sperm motility. The technical innovation is commendable and valuable for resource-limited laboratories. The operational definition of homogenization and the open availability of design files are important contributions.

However, the current manuscript in my opinion still requires major revisions. The main issues concern clarity of objectives, framing of the biological experiments, statistical transparency, and overemphasis on technical details. With revisions, this study could provide a meaningful resource for both bioengineering and reproductive biology.

Major comments

1. Clarity of objectives

The manuscript presents two parallel goals: (1) developing and validating the CASHo device, and (2) testing serotonin’s effects on sperm motility. At present, the serotonin experiment is insufficiently justified and feels appended.

Please clarify in the Introduction and Discussion whether the serotonin study is intended as a proof-of-concept demonstration of the device’s utility, or as a hypothesis-driven biological experiment. The framing will affect how results are interpreted.

Answer: Thank you for your valuable comment. We have clarified in the Introduction and Discussion that the serotonin experiment serves as a proof-of-concept demonstration of CASHo’s functionality, rather than a hypothesis-driven study. Corresponding sentences have been added to clearly frame the dual aims of the manuscript.

1.- In Introduction the objective 2 now reads: “As a proof-of-concept we use CASHo to investigate the effect of potential over-homogenization on the structure of kinematic subpopulations in sperm exposed to 5-HT in a non-capacitating medium.”

2.- In Discussion “In this study, the serotonin experiment was conceived as a functional demonstration of CASHo’s utility.”

2. Title and scope

The title (“Welcome to the open-hardware sperm quality lab…”) is catchy but misleading, since the work focuses on one device, not a full laboratory platform.

Consider a more precise title highlighting the homogenizer itself (e.g., “Development of an open-hardware semen homogenizer and application to serotonin effects on sperm motility”).

Answer: Thank you for your valuable comment. The title has been modified as suggested to: “Development of an open-hardware semen homogenizer and application to serotonin effects on sperm motility.”

3. Biological relevance and missing data

While motility subpopulation analysis is valuable, basic semen quality measures (total and progressive motility, morphology, integrity) should be reported alongside advanced kinematic parameters.

The absence of sperm morphology or integrity data limits assessment of the device’s impact on sperm quality. At minimum, please acknowledge this limitation more directly in the Discussion.

Answer: Thank you for your valuable comments. Total and progressive motility are presented in Figure 2. We have added a statement in the Discussion explicitly acknowledging the absence of sperm morphology and integrity data as a limitation of the study. The text readsThe text reads “Future research should investigate whether these factors influence sperm viability, membrane integrity, acrosome status, mitochondrial function, or DNA fragmentation, as their absence represents a limitation of the present study.”

4. Biological interpretation of clustering

While subpopulation analysis of sperm motility is an interesting approach, the manuscript does not fully explain what additional biological insights this provides compared to conventional mean values. For instance, the percentage shift (relative share) between subpopulations under different regimes and treatments should be visualized and discussed. This would clarify whether clustering reveals meaningful biological patterns (e.g., redistribution of sperm between more and less motile groups) beyond what is observed from average parameters.

At present, the added value of clustering remains unclear. Please expand the Discussion to explain what new information clustering contributes, how it should be interpreted biologically, and why it matters compared to traditional analyses.

Answer: Thank you for this insightful comment. Discussion was re-structured to highlight the advantage of subpopulation analysis over average analysis, mainly focused to the differential response of the kinematic subpopulation to 5-HT.

5. Comparisons with existing methods

The manuscript claims that commercial devices are unsuitable for gentle homogenization, but no direct experimental comparison is presented.

Please either:

a) provide direct comparative data (preferred), or

b) clearly state that claims are based on published studies and acknowledge the lack of head-to-head testing.

Answer: Thank you for your valuable comment. We have now clarified in the section “Limitations of existing commercial homogenization devices” that no direct experimental comparison with commercial equipment was performed. The statements regarding their limitations are explicitly based on previously published studies.

Specifically:

1.- The subheading was changed from “Effects of mixing using commercial devices” to “Limitations of existing commercial homogenization devices” to better reflect the scope of the discussion.

2.- The final paragraph of this section was rewritten to explicitly acknowledge that the discussion of commercial devices is drawn from prior reports and not from direct experimental testing.

3.- The following sentence was added at the end of the section:

“Although no direct comparison with commercial homogenizers was conducted, our discussion of existing devices and their reported effects is based on previously published studies. Therefore, claims regarding their limitations should be interpreted in this context, and future work should include direct comparative testing to validate these observations experimentally.”

These modifications ensure transparency about the evidence base supporting our statements and align the section with option (b) suggested by the reviewer.

6. Terminology and conceptual framing

The term homogenization is potentially misleading, as it usually refers to mechanical disruption of cells. Please clarify early in the text that the intended meaning is resuspension/mixing of spermatozoa and solutes.

Distinguish between different processes: (1) re-suspension of spermatozoa after sedimentation, (2) mixing of soluble additives such as serotonin, and (3) incubation for biological effects.

Answer: Thank you for your valuable comment and suggestions. The intended meaning of homogenization was clarified in Introduction, to read “Sample homogenization—mixing sufficiently to ensure consistent sperm cell counts is essential for semen evaluation, both in the production of insemination doses—to ensure consistent sperm cell counts—and in research applications, to obtain representative and reliable measurements.” In addition, the Material and methods section now explicitly distinguishes between three processes:

(1) resuspension of spermatozoa after sedimentation, now reads “During storage, sperm cells tend to sediment, making resuspension of spermatozoa is required prior to use.”

(2) mixing of soluble additives such as serotonin, now reads “...but incubation with 5-HT was still ongoing, requiring additional mixing of the additive.”

(3) incubation for biological effects, now reads “Sperm motility was evaluated immediately after each set of cycles.”

7. Over-homogenization

The concept of “over-homogenization” is central but underdeveloped. Please define it operationally (e.g., cycles beyond stabilization of sperm concentration) and explain its biological consequences.

Clarify whether this is species-specific (boar semen) or a generalizable phenomenon.

Answer: Thank you for your valuable suggestion.

1.- We have included the operational definition of over-homogenization in the subsection “Operational definition of objective homogenization and over-homogenization of semen samples.” The text now reads:

“A semen sample was considered over-homogenized when the number of mixing cycles applied in CASHo exceeded the level required to achieve homogenization.”

2.- The following clarification regarding the potential biological consequences of over-homogenization has been added to the Future directions:

“Because the number of cycles required to reach objective homogenization may vary across species, the threshold at which over-homogenization occurs is also likely to differ….Furthermore, studies across different species and extender formulations could clarify whether the impact of homogenization cycles and serotonergic signaling is generalizable or context-dependent.”

Answer: 8. Discussion writting

In several places, the Discussion begins with references to other studies rather than with the authors’ own findings. It is generally clearer to start each paragraph by restating the relevant result from the current study, and then relate it to existing literature. For example, in lines 557–565, the text does not clearly connect the study’s own data to the cited work. I recommend restructuring the Discussion so that the authors’ results are presented first, followed by comparison with previous findings (whether in agreement, contradiction, or expansion), along with possible explanations for similarities or differences. This will strengthen the logical flow and emphasize the study’s contribution.

Answer: Thank you for your valuable comment. The Discussion section has been revised and reorganized in several parts (including the section highlighted in your comment) to present our results first, followed by comparisons with previous studies. This modification improves the logical flow and emphasizes the contribution of the present study.

Minor comments

Organization and length

The revised structure (Methods following Introduction, supplementary hardware details moved to OSF) is much better. However, the main text is still too long (wordy). Consider further condensing hardware descriptions and focusing more on biological findings.

Answer: Thank you for your valuable suggestion. We have reviewed the hardware and operational descriptions once again and condensed them wherever possible. The remaining content was kept to the minimum necessary to ensure that the design and functioning of CASHo are comprehensible and reproducible. We believe this level of detail is essential for readers to fully understand the device and its validation.

1. Figures and tables

Some figures remain redundant or overly detailed. For example, raw individual variability plots may be better suited as Supplementary Materials, with main text showing summarized results.

In addition to reporting kinematic parameters, the manuscript could benefit from a visualization of how the relative proportion of sperm subpopulations shifts under different homogenization regimes and treatments. Presenting percentage shares (e.g., stacked bar plots, mosaic plots, or similar) would make these shifts more intuitive and highlight whether serotonin or over-homogenization leads to redistribution between motile and less motile clusters.

Answer: Thank you for your valuable comments.

1.- We are not entirely sure which specific figure you consider potentially redundant. Are you referring to Figure 4? This figu

---

## [Editor Report · Decision Letter 2]

21 Oct 2025

We look forward to receiving your revised manuscript.

Kind regards,

Sayyed Mohammad Hadi Alavi

Academic Editor

PLOS ONE

Journal Requirements:

Additional Editor Comments:

Thank you for submitting your manuscript to PLOS ONE. Please upload the highlighted MS showing details of revision. 

The highlighted MS showing revisions has not been uploaded. Please revise your submission.

---

## [Author Response · Author response to Decision Letter 3]

21 Oct 2025

The authors of the manuscript PONE-D-25-30420R2 “Welcome to the open-hardware sperm quality lab, featuring the new homogenizer: testing serotonin's effects on sperm motility” are grateful with Editor and reviewer's for their valuable time and suggestions to improve our manuscript. Each one of the comments and suggestions made by Editor and reviewers were addressed, as specified below. Please let me know any other comment or suggestion.

Best regards,

Dr. Andrés Aragón

Reviewer #1: The manuscript “Welcome to the open-hardware sperm quality lab, featuring the new homogenizer: testing serotonin's effects on sperm motility” is aims to design, construct, and validate an objective homogenizer for boar semen doses.

The revision substantially addresses the core reviewer concerns (design, modelling, structure, subpopulation analysis, clarity). The methodology is largely sound for a research article, with two moderate limitations: (i) absent morphology/functional markers and (ii) no empirical benchmarking vs. standard mixers. The statistical framework is appropriate overall, but I recommend (a) pairing/mixed modelling also at 5-cycle comparisons and (b) stating multiplicity handling or emphasizing effect sizes/CIs before acceptance.

I will recommend a moderate revision (address stats presentation at 5 cycles; add a brief multiplicity statement; explicitly flag morphology/functional markers as a limitation in Discussion; consider tightening the title). If these are implemented, I’d consider it publishable on the strength of its open-hardware contribution and transparent methodology

Answer: Thank you for your kind comments. We are grateful for your time and effort to help us to improve our manuscript. We hope that our responses and the corresponding modifications meet your expectations:

(i) We have added a statement in the Discussion explicitly acknowledging the absence of sperm morphology and integrity data as a limitation of the study. The text reads “Future research should investigate whether these factors influence sperm viability, membrane integrity, acrosome status, mitochondrial function, or DNA fragmentation, as their absence represents a limitation of the present study.”

(ii) We have added a statement in first paragraph of Discussion explicitly acknowledging the lacking of empirical benchmarking vs. standard mixers data. The text reads “CASHo represents a valuable alternative to commercial or open-hardware devices not specifically intended for semen homogenization. This assessment is based on technical information provided by manufacturers, as no direct experimental comparisons with commercial devices were performed in this study.”

(a) In the Statistics section, we originally wrote: “The effect of 5-HT at five cycles on total motility, progressive motility, and kinematic parameters (VCL, VAP, VSL, LIN, STR, WOB, BeatCross, and ALH) was compared using Student’s t-test,” because five cycles were considered the condition at which the sample was fully homogenized. Under this well-homogenized condition, we compared the Vh and 5-HT groups for total motility, progressive motility, and kinematic parameters. Subsequently, the five-cycle condition was used as the reference level in the mixed-effects model to analyze the over-homogenization conditions. To clarify we rephrased as follows “At five cycles, the effect of 5-HT treatment on total motility, progressive motility, and kinematic parameters (VCL, VAP, VSL, LIN, STR, WOB, BeatCross, and ALH) was assessed separately using a Student’s t-test, since this condition represents the point of complete homogenization and the comparison focused solely on treatment (5-HT vs. vehicle). Subsequently, the five-cycle condition was used as the reference level in the mixed-effects model, fitted to account for potential variability between boars, to evaluate over-homogenization effects across 10, 15, 20, and 25 cycles.”

(b) We thank the reviewer for this valuable suggestion. Because the aim of this study was exploratory rather than confirmatory, no formal multiplicity correction was applied. Instead, we placed emphasis on the magnitude and precision of the estimated effects, which are reported as effect sizes and 95% confidence intervals for fixed effects and interactions in S2 Table. This approach allows readers to evaluate the strength and direction of the observed effects beyond statistical significance.

Finally, the title was modified accordingly; it now reads “Development of an open-hardware semen homogenizer and application to serotonin effects on sperm motility.”

Reviewer #2: Manuscript title: Welcome to the open-hardware sperm quality lab, featuring the new homogenizer: testing serotonin's effects on sperm motility

Manuscript ID: PONE-D-25-30420_R1

Overall assessment

This manuscript presents the development and validation of an open-hardware semen homogenizer (CASHo) and its application in evaluating the effects of serotonin on boar sperm motility. The technical innovation is commendable and valuable for resource-limited laboratories. The operational definition of homogenization and the open availability of design files are important contributions.

However, the current manuscript in my opinion still requires major revisions. The main issues concern clarity of objectives, framing of the biological experiments, statistical transparency, and overemphasis on technical details. With revisions, this study could provide a meaningful resource for both bioengineering and reproductive biology.

Major comments

1. Clarity of objectives

The manuscript presents two parallel goals: (1) developing and validating the CASHo device, and (2) testing serotonin’s effects on sperm motility. At present, the serotonin experiment is insufficiently justified and feels appended.

Please clarify in the Introduction and Discussion whether the serotonin study is intended as a proof-of-concept demonstration of the device’s utility, or as a hypothesis-driven biological experiment. The framing will affect how results are interpreted.

Answer: Thank you for your valuable comment. We have clarified in the Introduction and Discussion that the serotonin experiment serves as a proof-of-concept demonstration of CASHo’s functionality, rather than a hypothesis-driven study. Corresponding sentences have been added to clearly frame the dual aims of the manuscript.

1.- In Introduction the objective 2 now reads: “As a proof-of-concept we use CASHo to investigate the effect of potential over-homogenization on the structure of kinematic subpopulations in sperm exposed to 5-HT in a non-capacitating medium.”

2.- In Discussion “In this study, the serotonin experiment was conceived as a functional demonstration of CASHo’s utility.”

2. Title and scope

The title (“Welcome to the open-hardware sperm quality lab…”) is catchy but misleading, since the work focuses on one device, not a full laboratory platform.

Consider a more precise title highlighting the homogenizer itself (e.g., “Development of an open-hardware semen homogenizer and application to serotonin effects on sperm motility”).

Answer: Thank you for your valuable comment. The title has been modified as suggested to: “Development of an open-hardware semen homogenizer and application to serotonin effects on sperm motility.”

3. Biological relevance and missing data

While motility subpopulation analysis is valuable, basic semen quality measures (total and progressive motility, morphology, integrity) should be reported alongside advanced kinematic parameters.

The absence of sperm morphology or integrity data limits assessment of the device’s impact on sperm quality. At minimum, please acknowledge this limitation more directly in the Discussion.

Answer: Thank you for your valuable comments. Total and progressive motility are presented in Figure 2. We have added a statement in the Discussion explicitly acknowledging the absence of sperm morphology and integrity data as a limitation of the study. The text readsThe text reads “Future research should investigate whether these factors influence sperm viability, membrane integrity, acrosome status, mitochondrial function, or DNA fragmentation, as their absence represents a limitation of the present study.”

4. Biological interpretation of clustering

While subpopulation analysis of sperm motility is an interesting approach, the manuscript does not fully explain what additional biological insights this provides compared to conventional mean values. For instance, the percentage shift (relative share) between subpopulations under different regimes and treatments should be visualized and discussed. This would clarify whether clustering reveals meaningful biological patterns (e.g., redistribution of sperm between more and less motile groups) beyond what is observed from average parameters.

At present, the added value of clustering remains unclear. Please expand the Discussion to explain what new information clustering contributes, how it should be interpreted biologically, and why it matters compared to traditional analyses.

Answer: Thank you for this insightful comment. Discussion was re-structured to highlight the advantage of subpopulation analysis over average analysis, mainly focused to the differential response of the kinematic subpopulation to 5-HT.

5. Comparisons with existing methods

The manuscript claims that commercial devices are unsuitable for gentle homogenization, but no direct experimental comparison is presented.

Please either:

a) provide direct comparative data (preferred), or

b) clearly state that claims are based on published studies and acknowledge the lack of head-to-head testing.

Answer: Thank you for your valuable comment. We have now clarified in the section “Limitations of existing commercial homogenization devices” that no direct experimental comparison with commercial equipment was performed. The statements regarding their limitations are explicitly based on previously published studies.

Specifically:

1.- The subheading was changed from “Effects of mixing using commercial devices” to “Limitations of existing commercial homogenization devices” to better reflect the scope of the discussion.

2.- The final paragraph of this section was rewritten to explicitly acknowledge that the discussion of commercial devices is drawn from prior reports and not from direct experimental testing.

3.- The following sentence was added at the end of the section:

“Although no direct comparison with commercial homogenizers was conducted, our discussion of existing devices and their reported effects is based on previously published studies. Therefore, claims regarding their limitations should be interpreted in this context, and future work should include direct comparative testing to validate these observations experimentally.”

These modifications ensure transparency about the evidence base supporting our statements and align the section with option (b) suggested by the reviewer.

6. Terminology and conceptual framing

The term homogenization is potentially misleading, as it usually refers to mechanical disruption of cells. Please clarify early in the text that the intended meaning is resuspension/mixing of spermatozoa and solutes.

Distinguish between different processes: (1) re-suspension of spermatozoa after sedimentation, (2) mixing of soluble additives such as serotonin, and (3) incubation for biological effects.

Answer: Thank you for your valuable comment and suggestions. The intended meaning of homogenization was clarified in Introduction, to read “Sample homogenization—mixing sufficiently to ensure consistent sperm cell counts is essential for semen evaluation, both in the production of insemination doses—to ensure consistent sperm cell counts—and in research applications, to obtain representative and reliable measurements.” In addition, the Material and methods section now explicitly distinguishes between three processes:

(1) resuspension of spermatozoa after sedimentation, now reads “During storage, sperm cells tend to sediment, making resuspension of spermatozoa is required prior to use.”

(2) mixing of soluble additives such as serotonin, now reads “...but incubation with 5-HT was still ongoing, requiring additional mixing of the additive.”

(3) incubation for biological effects, now reads “Sperm motility was evaluated immediately after each set of cycles.”

7. Over-homogenization

The concept of “over-homogenization” is central but underdeveloped. Please define it operationally (e.g., cycles beyond stabilization of sperm concentration) and explain its biological consequences.

Clarify whether this is species-specific (boar semen) or a generalizable phenomenon.

Answer: Thank you for your valuable suggestion.

1.- We have included the operational definition of over-homogenization in the subsection “Operational definition of objective homogenization and over-homogenization of semen samples.” The text now reads:

“A semen sample was considered over-homogenized when the number of mixing cycles applied in CASHo exceeded the level required to achieve homogenization.”

2.- The following clarification regarding the potential biological consequences of over-homogenization has been added to the Future directions:

“Because the number of cycles required to reach objective homogenization may vary across species, the threshold at which over-homogenization occurs is also likely to differ….Furthermore, studies across different species and extender formulations could clarify whether the impact of homogenization cycles and serotonergic signaling is generalizable or context-dependent.”

Answer: 8. Discussion writting

In several places, the Discussion begins with references to other studies rather than with the authors’ own findings. It is generally clearer to start each paragraph by restating the relevant result from the current study, and then relate it to existing literature. For example, in lines 557–565, the text does not clearly connect the study’s own data to the cited work. I recommend restructuring the Discussion so that the authors’ results are presented first, followed by comparison with previous findings (whether in agreement, contradiction, or expansion), along with possible explanations for similarities or differences. This will strengthen the logical flow and emphasize the study’s contribution.

Answer: Thank you for your valuable comment. The Discussion section has been revised and reorganized in several parts (including the section highlighted in your comment) to present our results first, followed by comparisons with previous studies. This modification improves the logical flow and emphasizes the contribution of the present study.

Minor comments

Organization and length

The revised structure (Methods following Introduction, supplementary hardware details moved to OSF) is much better. However, the main text is still too long (wordy). Consider further condensing hardware descriptions and focusing more on biological findings.

Answer: Thank you for your valuable suggestion. We have reviewed the hardware and operational descriptions once again and condensed them wherever possible. The remaining content was kept to the minimum necessary to ensure that the design and functioning of CASHo are comprehensible and reproducible. We believe this level of detail is essential for readers to fully understand the device and its validation.

1. Figures and tables

Some figures remain redundant or overly detailed. For example, raw individual variability plots may be better suited as Supplementary Materials, with main text showing summarized results.

In addition to reporting kinematic parameters, the manuscript could benefit from a visualization of how the relative proportion of sperm subpopulations shifts under different homogenization regimes and treatments. Presenting percentage shares (e.g., stacked bar plots, mosaic plots, or similar) would make these shifts more intuitive and highlight whether serotonin or over-homogenization leads to redistribution between motile and less motile clusters.

Answer: Thank you for your valuable comments.

1.- We are not entirely sure which specific figure you consider potentially redundant. Are you referring to Figure 4? This fi

---

## [Editor Report · Decision Letter 3]

6 Nov 2025

Dear Dr. Aragón,

We look forward to receiving your revised manuscript.

Kind regards,

Sayyed Mohammad Hadi Alavi

Academic Editor

PLOS ONE

Journal Requirements:

Additional Editor Comments:

Dear Prof. Andrés Aragón

Thank you very much for your submission. I have carefully reviewed revised version of your MS as well as your point-by-point responses to the reviewer’s comments. Hence the MS quality improved compared to the original submission, however it is not suitable for publication in its current version. Therefore, I would like to ask you another, BUT THE LAST REVISION, to reach a publishable version. Here, I am providing you with an eight weeks due time, if it needs more, please let me know. I would very much appreciate it if you consider one-by-one comment/recommendation/question, and revise the MS.

Very importantly;

1- Please include, a brief description of CASHo and operational information in the supplementary materials, including components that has been provided in the original submission.

2- The most comments from all three reviewers corresponds to the statistical analyses. I have also a big concern. I would appreciate if you double check with following introduction and guide, and revise the MS. For more details, please read the following key reference: Seltman, H. J. 2018. “Two‐Way ANOVA.” In Experimental Design and Analysis, 267–292. Carnegie Mellon University Publication. https://
www.stat.cmu.edu/~hseltman/309/Book/Book.pdf.

The statistical analysis has used two-way or repeated-measure ANOVA; however the method for operation this kind of analysis needs revision.

In the present study, there are two main factors (homogenization cycle and 5-HT). To conduct statistical analysis, very simply apply a two-way ANOVA and investigate the effects of homogenization cycle, 5-HT, and their interactions (homogenization cycle x 5-HT). THIS HAS BEEN DONE. However, when interaction effects of 5-HT x cycle is significant for a parameter (such as VCL, VSL, … in your case), then it is enough to conduct one-way ANOVA and compare each parameter between 5-HT and vehicle AT EACH CYCLE (We already know that 5 cycle is well enough for sperm homogenization). To better get familiar with this kind of analysis; I have included two articles form our works that provide step-by-step stat procedures. This is not to cite these works as references, I am not going to receive bias citation. It is just for your kind considerations to revise the MS correctly, and present the findings in a correct form.

(1) https://doi.org/10.1371/journal.pone.0243569

(2) https://doi.org/10.1002/jez.2918

3- There are a large numbers of comments/recommendations to your MS that I have included directly in the file using a track-change. I would appreciate if you kindely and carefully double check them and revise the MS to reach an acceptable version for publication.

4- If you follow comments no. 2; you may find that Supplementary tables 1 and 2 may not be necessary.

5- To my experience, I could not access the video files you uploaded on OSF. I would suggest to support your submission with adding the videos into supplementary materials of your MS.
---

## [Author Response · Author response to Decision Letter 4]

19 Nov 2025

The authors of the manuscript PONE-D-25-30420R3 “Development of an open-hardware semen homogenizer and application to serotonin effects on sperm motility” are grateful with Editor. Each one of the comments and suggestions were addressed, as specified below. We hope this version of our manuscript meet the standards of quality of PLOSONE.

Best regards,

Dr. Andrés Aragón

Very importantly;

1- Please include, a brief description of CASHo and operational information in the supplementary materials, including components that has been provided in the original submission.

Answer. Thank you for your recommendation. The supplementary file 1 now include description of CASHo, operational information and components as in the original submission.

2- The most comments from all three reviewers corresponds to the statistical analyses. I have also a big concern. I would appreciate if you double check with following introduction and guide, and revise the MS. For more details, please read the following key reference: Seltman, H. J. 2018. “Two‐Way ANOVA.” In Experimental Design and Analysis, 267–292. Carnegie Mellon University Publication. https://
www.stat.cmu.edu/~hseltman/309/Book/Book.pdf.

The statistical analysis has used two-way or repeated-measure ANOVA; however the method for operation this kind of analysis needs revision.

In the present study, there are two main factors (homogenization cycle and 5-HT). To conduct statistical analysis, very simply apply a two-way ANOVA and investigate the effects of homogenization cycle, 5-HT, and their interactions (homogenization cycle x 5-HT). THIS HAS BEEN DONE. However, when interaction effects of 5-HT x cycle is significant for a parameter (such as VCL, VSL, … in your case), then it is enough to conduct one-way ANOVA and compare each parameter between 5-HT and vehicle AT EACH CYCLE (We already know that 5 cycle is well enough for sperm homogenization). To better get familiar with this kind of analysis; I have included two articles form our works that provide step-by-step stat procedures. This is not to cite these works as references, I am not going to receive bias citation. It is just for your kind considerations to revise the MS correctly, and present the findings in a correct form.

(1) https://doi.org/10.1371/journal.pone.0243569

(2) https://doi.org/10.1002/jez.2918

Answer. We are grateful for your comments and for the references you provided. We reviewed them carefully and revised our statistical procedure accordingly. In the studies you referred to, two explanatory variables with multiple levels were evaluated across different parameters. In our case, we also have two explanatory variables: Treatment with two levels (Vh and 5-HT) and Cycles with five levels (5, 10, 15, 20, and 25). Therefore, for a given motility parameter, when the interaction term in the linear mixed model was significant, we decomposed the analysis by performing one-way ANOVAs to compare the means of the Vh and 5-HT groups at each cycle level.

We incorporated the following text into the Statistics section: “When a significant interaction was detected, the models were further analyzed by performing separate one-way ANOVAs at each cycle level. Conversely, if the interaction was not significant, the main effects of cycles and 5-HT were evaluated directly.”

We also modified the figures containing interaction plots by adding asterisks to indicate differences between the Vh and 5-HT groups at each cycle level. The corresponding figure legends were updated accordingly.

3- There are a large numbers of comments/recommendations to your MS that I have included directly in the file using a track-change. I would appreciate if you kindely and carefully double check them and revise the MS to reach an acceptable version for publication.

Answer. Thank you for your helpful comments and recommendations on the manuscript; each of them was carefully addressed. The revised manuscript containing your suggestions has been uploaded as the “Revised Manuscript with Track Changes.” All comments are now marked as “Resolved” in LibreOffice. In some cases, when text was deleted, the comment appears crossed out, but it was nevertheless fully addressed. In the PDF version generated by the PLOS ONE platform, the comments appear with different shading.

4- If you follow comments no. 2; you may find that Supplementary tables 1 and 2 may not be necessary.

Answer. Thank you for your comment. We addressed the issues raised in points 1 and 2; therefore, Supplementary Tables 1 and 2 were deleted. The S1 File now contains the description, operational information, and components of CASHo. In accordance with point 2, S2 Table was also removed.

5- To my experience, I could not access the video files you uploaded on OSF. I would suggest to support your submission with adding the videos into supplementary materials of your MS.

Answer. Thank you for your kind comment. You are right—the videos cannot be viewed online on OSF. I have reported this issue to the OSF administrator, and their engineers are currently working to resolve it. In the meantime, I have added the following note to the “Videos” section of the project Wiki: “If the videos cannot be viewed online on OSF, please download them.”

We have also added Videos S1 and S2 to the supplementary materials of the manuscript.

---

## [Editor Report · Decision Letter 4]

23 Nov 2025

Development of an open-hardware semen homogenizer and application to serotonin effects on sperm motility

PONE-D-25-30420R4

Dear Dr. Aragón,

We’re pleased to inform you that your manuscript has been judged scientifically suitable for publication and will be formally accepted for publication once it meets all outstanding technical requirements.

Kind regards,

Sayyed Mohammad Hadi Alavi

Academic Editor

PLOS ONE

Additional Editor Comments (optional):

Dear Dr. Andrés Aragón

Thank you for your revision. The MS is now well constructed. I am glad to inform you that the MS has been accepted for publication. However, when I read the MS once again, I found a few number of small errors that could be omitted. Please contact Publication office, and ask for a final reading before being subjected to page-setting for publication. For instance, L368-371 and L372-274 are fully similar. One should be deleted.

Very best regards

Hadi Alavi, AE, PLOS ONE
---

## [Editor Report · Acceptance letter]

PONE-D-25-30420R4

PLOS ONE

Dear Dr. Aragón,

I'm pleased to inform you that your manuscript has been deemed suitable for publication in PLOS ONE. Congratulations! Your manuscript is now being handed over to our production team.

Kind regards,

on behalf of

Dr. Sayyed Mohammad Hadi Alavi

Academic Editor

PLOS ONE